# Engineering thin 3D Li-composite foil negative electrodes with high mechanical toughness

Yu-Hao Wang[1,2,3], Shuang-Jie Tan [1,3], Chao-Hui Zhang[1], Jun-Chen Guo[1], Xiao-Xi Luo[1], Ruo-Xi Jin[1,2], Lin-Bo Huang[1], Xiao-Chuan Su[1,2], Chen Li[1,2], Xu-Sheng Zhang [1], Xing Zhang[1], Sen Xin [1,2], Rui Wen [1,2], Juan Zhang[1] ✉ & Yu-Guo Guo [1,2] ✉

Current three-dimensional lithium negative electrodes are plagued by inherent trade-offs among mechanical robustness, thin processability, and electrochemical performance. Here, we engineer a free-standing Li-composite foil negative electrodes by integrating a lithiophilic Li-Zn alloy with a $Li_3N$-enriched carbon nanotube network. The Li-Zn alloy strengthens tensile resistance and regulates lithium deposition, while the $Li_3N$-enriched carbon nanotube network reinforces mechanical toughness, achieving a rupture toughness of $1.3 \times 10^6$ $J/m^3$, a 12-fold enhancement over bare lithium. This property enables the fabrication of thin negative electrodes (<10 μm) that resist pulverization during deep Li plating/stripping. In cells with $LiNi_{0.8}Co_{0.1}Mn_{0.1}O_2$ positive electrodes, the composite negative electrode facilitates extended cyclability (>500 cycles in coin cells at 1 C, 92% retention after 300 cycles in Ah-grade pouch cells at 0.5 C) and sustain high-rate operation (10 C). An 8.5 Ah pouch cell demonstrates a practical specific energy of 553 Wh $kg^{-1}$ at cell level when tested at 0.1 C. This work presents a design strategy for realizing high-energy, long-cycle-life lithium metal batteries.

Lithium metal has been regarded as the holy grail of the negative electrode for next-generation high-specific-energy batteries because of its high capacity (3860 mA h $g^{-1}$) and the lowest electrochemical potential (−3.04 V vs. standard hydrogen electrode)[1–5]. However, dendritic growth, low coulombic efficiency (CE) and severe volume fluctuations during cycling critically impede the practical implementation of lithium metal batteries (LMBs)[6–9]. Current strategies focus on electrolyte optimization[10–13], artificial interphase engineering[14–18], and solid-state electrolyte design[19–21]. Though these approaches demonstrated effectiveness in suppressing dendrite propagation[22–25], the volume fluctuations problems in planar lithium configurations remain as a critical barrier to practical implementation[26,27].

Infiltrating lithium into a three-dimensional (3D) host to construct composite lithium negative electrodes, a strategy pioneered by our group, has emerged as a promising approach to mitigate the challenges associated with volumetric changes. The 3D framework facilitates a homogeneous electric field distribution, thereby promoting uniform lithium deposition and mitigating the expansion during cycling[28]. This innovation has spurred extensive research into metallic (e.g., Cu and Ni)[29–31] and carbon-based 3D composite lithium negative electrodes[32–34]. Despite their advantages, the fabrication and processing of 3D Li-composite negative electrodes remain formidable challenges[35,36]. The carbon-based hosts suffer from poor lithophilicity, which inhibits complete lithium infiltration and results in undesirable inhomogeneity within the composite. Additionally, integrating a highly

[1]CAS Key Laboratory of Molecular Nanostructure and Nanotechnology/ Institute of Chemistry, Chinese Academy of Sciences (CAS), Beijing, P. R. China. [2]School of Chemical Sciences, University of Chinese Academy of Sciences (UCAS), Beijing, P. R. China. [3]These authors contributed equally: Yu-Hao Wang, Shuang-Jie Tan. ✉e-mail: zhangjuan120@iccas.ac.cn; ygguo@iccas.ac.cn

conductive ionic network within the 3D carbon architecture remains a critical challenge, restricting high-rate performance for these 3D Li–C negative electrodes. More importantly, the practical development of thin lithium metal negative electrodes (e.g., <50 μm) is essential to achieve a favorable negative-to-positive (N/P) ratio compatible with commercial positive electrodes. However, the fabrication of thin 3D Li–C negative electrodes remains hindered by processing limitations and inherent brittleness, further complicating their viability for large-scale implementation (Supplementary Fig. 1)[37,38]. Recent studies have explored an alternative approach of 3D Li-alloy foil negative electrodes via the reaction of molten lithium with metals or metal compounds at elevated temperatures. The incorporation of $Li_xM_y$ (M = Zn, Sn, Mg, etc.) alloy phases enhances the processability of this strategy. These alloy phases not only function as mechanical pillars, but also serve as lithophilic active sites that regulate lithium deposition (Supplementary Fig. 1)[39,40]. Nevertheless, the discontinuous alloy foil networks are prone to fracture during prolonged cycling, particularly in lithium-limited systems. Under these conditions, lithium depletion triggers dealloying-driven electrode pulverization, ultimately accelerating battery failure[41,42]. To mitigate this, 3D Li-alloy negative electrodes are often co-rolled with Cu or stainless-steel substrates, though at the expense of gravimetric capacity[43,44].

These unresolved challenges expose the fundamental dilemma in the design of 3D Li-composite negative electrode, which fails to simultaneously satisfy six critical requirements: (i) rapid ion/electron transport, (ii) scalable processing, (iii) mechanical toughness, (iv) minimal volume variation, (v) high specific energy, and (vi) structure integrity. Despite recent progress, the simultaneous optimization of these interdependent properties remains a pivotal scientific challenge that hinders the commercial realization of LMBs.

Herein, we present a 3D Li-composite foil negative electrode (LZNC) that achieves simultaneous multidimensional performance enhancement through synergistically integrating Li–Zn alloy with $Li_3N$-enriched carbon nanotubes ($Li_3N$ -CNTs). The Li–Zn alloy provides intrinsic ductility, effectively accommodating cyclic strain while regulating lithium nucleation. The $Li_3N$ -CNTs network acts as a mechanically durable scaffold, preserving structural integrity even under prolonged cycling or deep delithiation. This hierarchical integration results in a tensile strength of 23.9 MPa and a rupture toughness of $1.3 \times 10^6$ J/m³, approximately 1200% enhancement compared to bare Li, enabling the fabrication of thin composite foil negative electrodes (<10 μm). This thin yet robust 50 μm-thick LZNC foil delivers a high gravimetric specific capacity (1800 mA h g⁻¹) and a high volumetric capacity (1915 mA h cm⁻³), improving those 3D Li–C or Li-alloy composite negative electrodes. In addition, coin full cells pairing thin LZNC with $LiNi_{0.8}Co_{0.1}Mn_{0.1}O_2$ (NCM811) show stable cycling over 500 cycles and maintain good rate capability. LZNC‖NCM811 pouch cells demonstrate long-term performance under practical testing conditions (300 cycles with 92% capacity retention). A practical 8.5 Ah prototype LMBs pouch cell was constructed, demonstrating a high specific energy of 553 Wh kg⁻¹ at the cell-level, including the weight of the pouch packaging (aluminum-plastic film and tabs).

## Results

The fabrication procedure for the LZNC negative electrode is illustrated in Fig. 1a. First, molten lithium was reacted with $Zn_3N_2$ (mass ratio: 2:1) at 400 °C under an Ar atmosphere. Scanning electron microscopy (SEM) and transmission electron microscopy (TEM) confirm the uniform dispersion of $Zn_3N_2$ (Supplementary Figs. 2–4). Upon a thorough stirring, a redox chemical reaction between metallic Li and $Zn_3N_2$ occurs, yielding a hybrid matrix (LZN) comprising Li–Zn alloy and $Li_3N$. Notably, the in situ produced $Li_3N$, a fast ionic conductor, facilitates rapid lithium-ion transport throughout the 3D framework, as corroborated by X-ray diffraction (XRD) (Supplementary Fig. 5, Supplementary Table 1). Density functional theory (DFT)-derived ternary-

phase diagrams further validated the thermodynamic stability of LZNC at ambient conditions (Supplementary Fig. 6). To reinforce structural integrity and flexibility, CNTs were integrated into LZN, serving as bridging scaffolds. Unlike conventional systems where lithiophobic CNTs resist homogeneous dispersion, the LZN enables spontaneous CNTs wetting. This enhanced compatibility arises from $Li_3N$ enrichment on CNTs surfaces during annealing, a mechanism detailed in subsequent sections. The $Zn_3N_2$-to-CNTs ratio was systematically optimized to balance mechanical robustness and ionic conductivity (Supplementary Fig. 7).

The freestanding LZNC foil negative electrode was fabricated through ambient cooling of a molten Li–$Zn_3N_2$ precursor, followed by roll-pressing under dry air. In the development of LMBs, thin lithium negative electrodes (<20 μm) are essential for realizing high-specific-energy cells with minimized N/P ratios. Conventional fabrication of large-area or thin lithium foils, however, is challenged by their intrinsic limitations of high viscosity and insufficient mechanical strength. Addressing this, our LZNC design enables scalable production of thin foil negative electrodes (<20 μm). We further conducted complementary tensile tests on these foils to evaluate their mechanical properties (Fig. 1b, Supplementary Fig. S8). Quantitatively, the LZN composite exhibits a ~200% increase in tensile strength over the LZ baseline, an effect we ascribe to conventional second-phase reinforcement by the inorganic $Li_3N$ particles. The Li–Zn-CNTs (LZC composite shows an even greater strength enhancement of ~320%. Furthermore, we calculated the toughness, defined as the energy absorbed prior to fracture (integral under the stress-strain curve), for each material. While LZN shows a ~100% increase over LZ, LZC achieves ~240% enhancement. The synergistic combination of both components in the LZNC composite yields the highest toughness value ($1.3 \times 10^6$ J/m³, 300% enhancement to LZ). These data demonstrate that the CNTs network serves as the principal mechanical reinforcement, drastically improving both tensile strength and toughness by acting as a resilient scaffold that bridges Li–Zn domains, inhibits crack propagation, and effectively redistributes mechanical stress.

Atomic force microscopy (AFM) adhesion measurements further validate the enhanced mechanical processability of LZNC foil (Supplementary Fig. 9). By modulating roller spacing during fabrication, the LZNC foil thickness can be precisely tailored from 8 to 100 μm, enabling tunable areal capacities ranging from 1.5 to 19.15 mA h cm⁻² (Fig. 1c, Supplementary Fig. 10). The thin LZNC foils (8 μm) retain structural integrity under mechanical stressors such as bending, stretching, and folding, exhibiting no detectable cracks, pinholes, or defects (Fig. 1d, Supplementary Figs. 11, 12). This high mechanical resilience, coupled with flexibility, renders LZNC a promising negative electrode for operation under mechanical conditions. Furthermore, the composite foil demonstrates scalability, as evidenced by its successful processing into smooth, large-area foils (80 cm², 40 μm thickness; Fig. 1d), highlighting its viability for industrial-scale negative electrode production in next-generation LMBs. The actual specific capacity of LZNC negative electrode was determined by electrochemical charging to 1 V (vs. Li⁺/Li), revealing gravimetric and volumetric capacities of 1800 mA h g⁻¹ and 1915 mA h cm⁻³, respectively. The main components of LZNC composite materials are shown in Supplementary Table 2. The theoretical specific capacity is 1842 mA h g⁻¹, which is very close to the actual value, demonstrating high utilization of active lithium. The negative electrode demonstrated remarkably limited thickness variation (3.6%) upon complete lithium stripping (Fig. 1e), confirming its minimized volumetric change. Compared with prior studies on Li–C or Li-alloy 3D negative electrodes, the LZNC architecture provides higher gravimetric and volumetric capacities while also allowing for thin-electrode fabrication (Fig. 1f). To comprehensively evaluate the multifunctional advantages of LZNC, we conducted a comparative analysis against conventional lithium, Li–C, and Li-alloy negative electrodes using a six-parameter radar chart

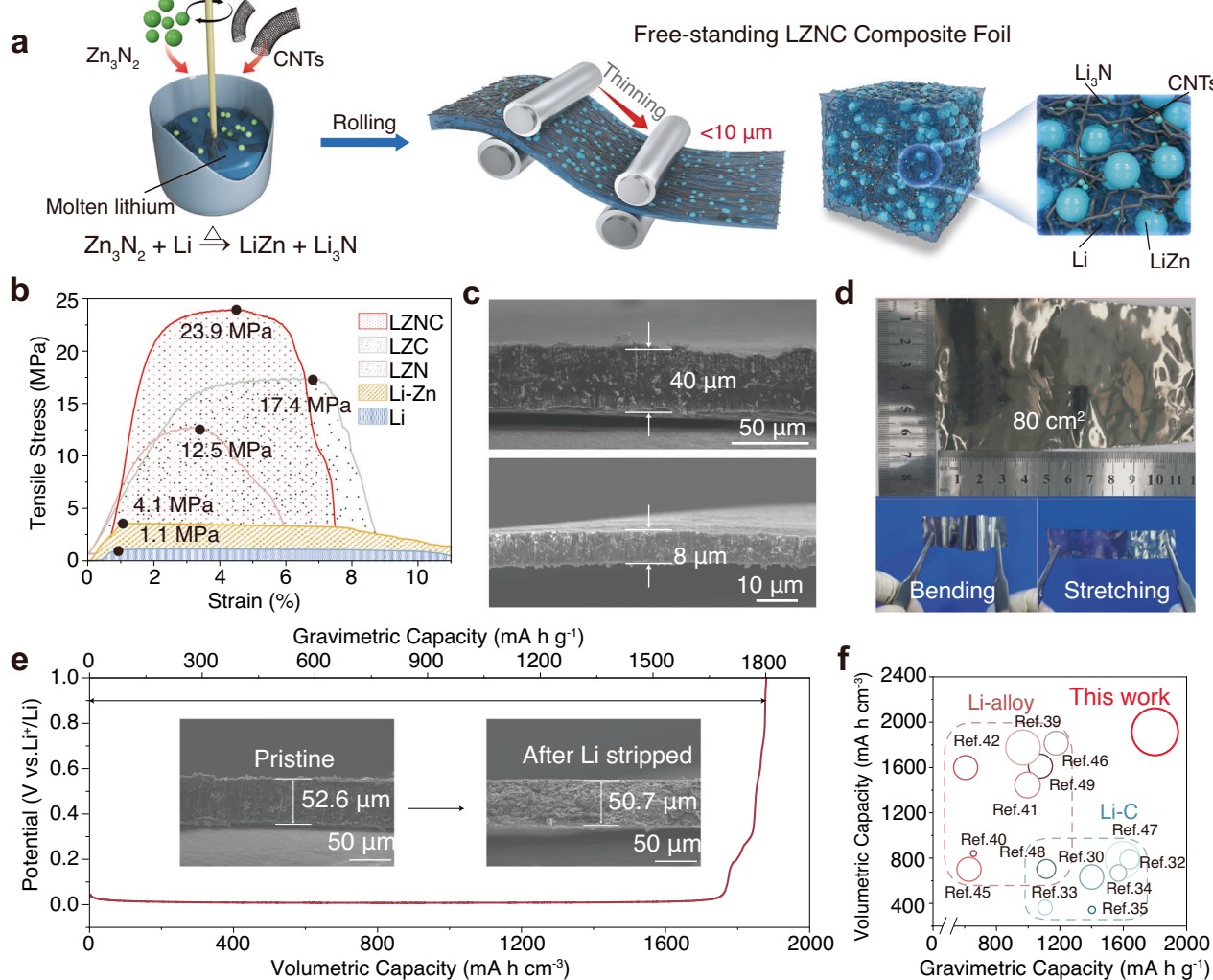

**Fig. 1 | Structural design, fabrication, and key properties of the LZNC composite anode. a** Schematic illustration of the fabrication process and architecture of the LZNC. **b** Tensile stress-displacement profiles of Li foil and the LZNC foil, with the integrated area beneath the stress-strain curve (shaded area) quantifying its fracture toughness. **c** Cross-sectional SEM images of LZNC with varying thicknesses. **d** Digital photographs of a 50 μm-thick LZNC foil, prepared over a large area, exhibiting mechanical toughness and flexibility. **e** Li stripping curve of the LZNC electrode charged to 1 V versus Li⁺/Li. Inset: the cross-sectional SEM images for LZNC before and after Li stripped. **f** Comparison of volumetric/gravimetric specific capacity of the previously reported 3D Li−C and 3D Li-alloy negative electrodes. Note that the circumferential area inversely correlates with the thickness of the composite negative electrodes[30,32-35,39-42,53-57]. Detailed parameters are provided in Supplementary Table 3.

framework. This analysis quantified performance across critical metrics: ion/electron transport, mechanical toughness, processability, volumetric deformation, gravimetric specific energy, and structural durability. The LZNC design synergistically integrates the structural advantages of Li−C and Li-alloy systems through a ternary-phase LZNC framework that suppresses volume fluctuations, enabling rapid charge transfer, and delivering robust mechanical resilience/scalability, thereby addressing the intrinsic brittleness of Li-alloys and processing limitations of Li−C to reconcile specific energy, structural integrity, and manufacturability trade-offs.

The structural and compositional evolution of the LZNC was thoroughly investigated. XRD results (Fig. 2a) confirm the complete conversion of $Zn_3N_2$ into Li−Zn alloy and $Li_3N$ during chemical reduction, as evidenced by the disappearance of $Zn_3N_2$ diffraction peaks and the emergence of characteristic LiZn, $Li_3N$, and metallic Li phases. Meanwhile, the lithiation of CNTs during thermal processing is evidenced by a 2 theta downshift and intensity reduction of their (002) diffraction peak (Supplementary Fig. 13). Detailed XRD and X-ray photoelectron spectroscopy (XPS) analyses further confirm the formation of lithium-intercalated carbon ($LiC_x$). The (002) peak shifts

from 25.92° to 23.85°, corresponding to an interlayer expansion from 3.57 to 3.73 Å, characteristic of Li insertion into graphitic layers. The relatively weak diffraction intensity originates from the low crystallinity of the lithiated carbon compared with the metallic phases. Moreover, the appearance of a Li−C bonding signal at 282.9 eV in the C 1s XPS spectrum provides direct chemical evidence for $LiC_x$ formation. Ex situ SEM observation (Supplementary Fig. 14) further reveals that the CNTs retain their tubular and interconnected morphology after lithiation, confirming that Li intercalation expands the lattice while preserving the robust fibrous framework. XPS validates these phase transformations, with distinct N 1s (398.9 eV) and Zn 2p (1044.4/1021.4 eV) binding energies corresponding to $Li_3N$ and LiZn alloy formation (Supplementary Fig. 15). The structural and compositional evolution of the LZNC was further elucidated using SEM, energy-dispersive X-ray spectroscopy (EDS), and X-ray microscopy (XRM). Top-view SEM images of the LZNC foil reveal that the LiZn exhibits an island-like morphology (~1 μm), while the CNTs form an interwoven network that bridges adjacent alloy clusters. Cross-sectional SEM and EDS analysis demonstrate that Li−Zn alloy particles, CNTs, and $Li_3N$ are uniformly dispersed throughout the foils (Supplementary Figs. 16, 17).

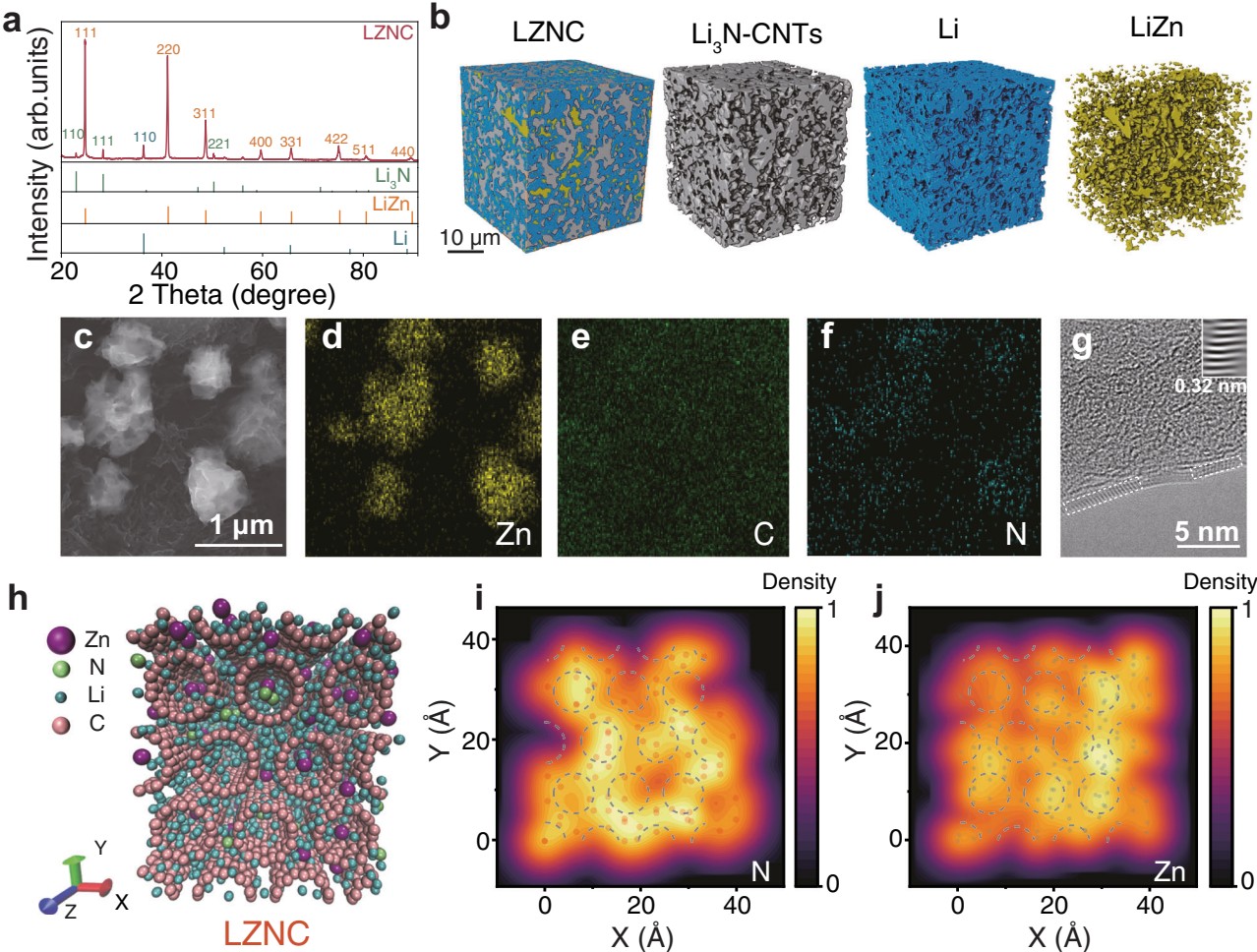

**Fig. 2 | Structural features and component distribution of the LZNC composite.** **a** XRD pattern of LZNC composite. **b** Distribution of CNTs/Li₃N and LiZn within the LZNC detected by XRM. **c** SEM image of LZNC after Li stripping at 0.5 mA cm⁻² with a cutoff voltage of 1 V vs Li⁺/Li and the corresponding element mapping for **d** Zn, **e** C, and **f** N. **g** TEM image of LZNC after Li stripping under the same conditions. **h** Schematic of the molecular dynamics simulation of LZNC after annealing. Atomic density mapping for **i** N and **j** Zn, with yellow regions indicating the highest probability of atomic presence, the dashed part represents the position of CNTs.

XRM further validates the presence of an interconnected 3D alloy-CNTs framework embedded in the lithium matrix (Fig. 2b, Supplementary Fig. 18). Even after complete lithium stripping, Zn particles remain interconnected with CNTs, maintaining good structural integrity (Fig. 2c–f). Intriguingly, the signals of N element almost overlap with those of C element in EDS mappings (Fig. 2e, f), suggesting strong interfacial coupling between Li₃N and the CNTs. This is further corroborated by TEM, which shows Li₃N crystallites preferentially enriched on CNTs surfaces with a lattice spacing of 0.32 nm, corresponding to the (111) plane (Fig. 2g). Molecular dynamics (MD) simulations were performed to investigate the dynamic evolution of the compositional distribution during the synthesis of LZNC. It reveals that during the thermal fusion process, N and Zn atoms are uniformly dispersed (Supplementary Fig. 19). After annealing, N atoms migrate preferentially to CNTs interfaces due to electronic structure mismatches between metallic and inorganic phases, while Zn atoms maintain homogeneous dispersion (Fig. 2h–j). This self-evolving architecture, where CNTs-bound Li₃N enhances ion transport kinetics within the LZNC. Further computational details are provided in Supplementary Data 1.

To gain a comprehensive understanding of the Li plating/stripping mechanism on the LZNC negative electrode, we employed a series of in situ and quasi-in situ characterizations. Operando optical microscopy (Supplementary Fig. 20) provides direct visualization of Li deposition and stripping at 1 mA cm⁻² with a capacity of 2 mA h cm⁻². The LZNC electrode retains a continuous and intact 3D skeleton throughout Li stripping, and subsequent Li plating occurs homogeneously within the scaffold without dendrite formation. In contrast, both Li–Zn alloy and bare Li electrodes exhibit evident structural degradation during stripping, including cracking and loosening of the framework, and irregular, dendritic Li deposition during plating. To further elucidate the structural and phase evolution during Li stripping and plating, in situ XRD measurements were conducted on the LZNC negative electrode. Figure 3a shows the in situ XRD profiles of LZNC during the first Li plating/stripping. Electrochemical testing was performed at a current density of 0.1 mA cm⁻², with delithiation first at a cut-off voltage of 1 V and then lithiation at a cut-off capacity of 3 mA h cm⁻². The XRD pattern shows that the peak strength and peak position of Li₃N remain unchanged throughout the Li stripping/plating process, indicating its electrochemical stability upon cycling. During Li stripping, the peak intensity of lithium gradually diminishes without peak shifts. Towards the end of the Li stripping (charge to -0.15 V), the voltage gradually rises while the peak intensity of LiZn diminishes, accompanied by the emergence of the peak of Zn. This indicates that LZNC undergoes a sequential Li-stripping and de-alloying mechanism during delithiation. This interface evolution is further corroborated by quasi-in situ XPS analysis (Fig. 3b), which reveals that the characteristic signals of Li₃N and metallic Zn remain essentially unchanged at

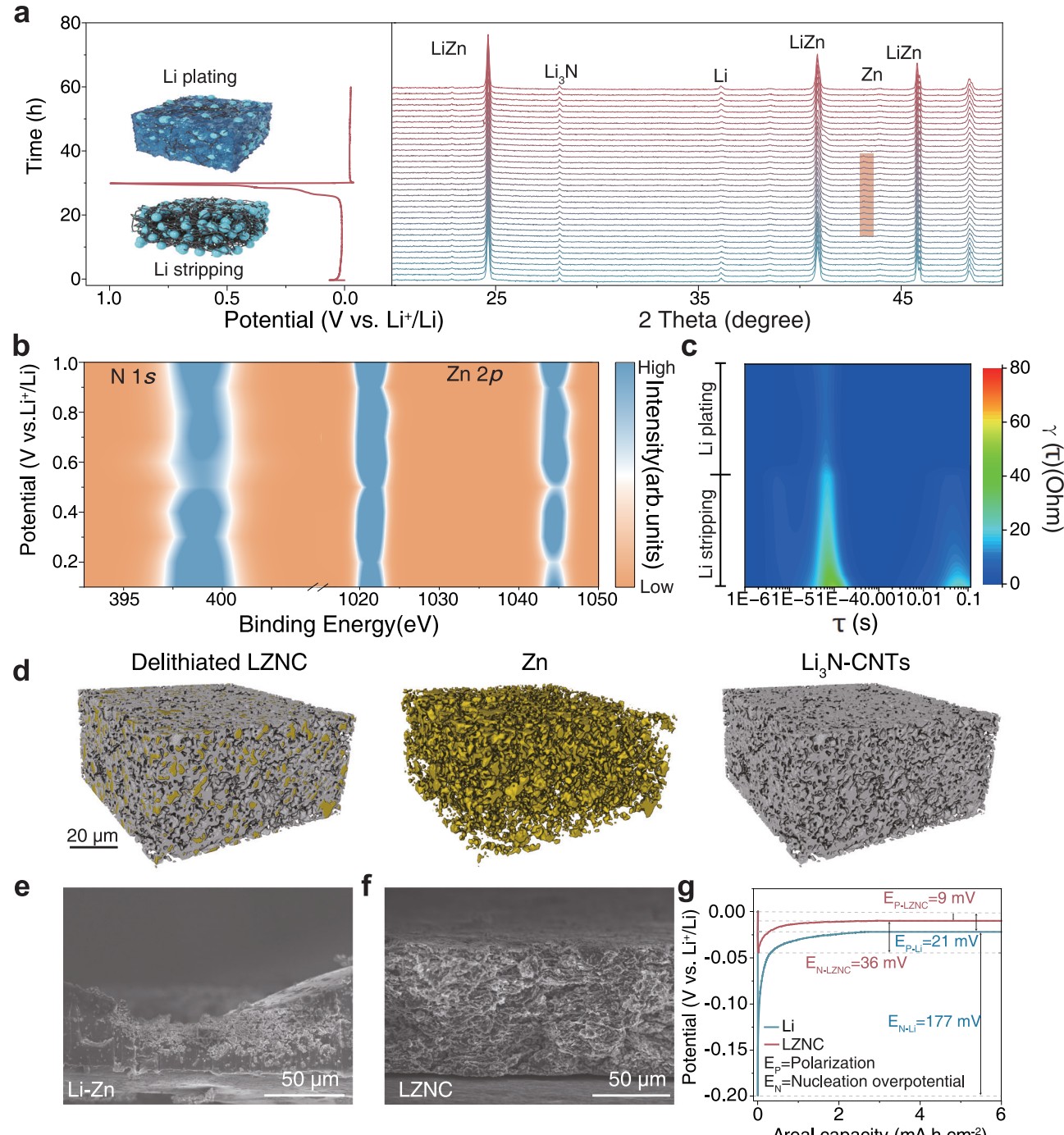

**Fig. 3 | In situ/ analyses revealing the Li stripping and plating mechanisms of the LZNC negative electrode. a** In situ XRD analysis and schematic illustration of the Li stripping and plating at 0.1 mA cm⁻² processes of LZNC negative electrode. **b** Quasi-in situ XPS analysis of changes in N 1 s and Zn 2p during Li stripping of LZNC negative electrodes at 0.5 mA cm⁻². **c** In situ DRT analysis of the Li stripping and plating at 0.5 mA cm⁻² processes of LZNC negative electrode. **d** X-ray CT reconstruction of the fully delithiated LZNC negative electrode after Li stripping at 0.5 mA cm⁻² with a cutoff voltage of 1 V vs Li⁺/Li. SEM images of **e** Li−Zn alloy and **f** LZNC after stripping of 75% free Li at 0.5 mA cm⁻². **g** Voltage profiles of 25% Li-stripped bare Li and LZNC negative electrodes with Li plating at 0.5 mA cm⁻²/6 mA h cm⁻².

different delithiation states, confirming the chemical robustness of the interface and supporting the in situ XRD observations. In situ Distribution of Relaxation Times (DRT) (Fig. 3c, Supplementary Fig. 21) reveals that the interfacial resistance of LZNC gradually decreases and stabilizes, indicating efficient charge transfer and a persistent interface, while Li−Zn exhibits large fluctuations due to framework collapse and dendritic deposition. Ex situ SEM and optical microscopy of Li−Zn and LZNC negative electrodes at 25% and 75% delithiation were

characterized (Fig. 3e, f, Supplementary Fig. 22). It reveals that the Li−Zn negative electrode suffers rapid structural collapse and electrode pulverization at 25% Li removal, whereas LZNC retains 3D framework integrity even at 75% delithiation. To further verify the structural reversibility after complete delithiation, ex situ SEM and EDS mapping were performed on an LZNC sample charged to 1.0 V (vs. Li⁺/Li). The results (Supplementary Fig. 23) show a continuous, interwoven network in which Zn domains display an island-like morphology

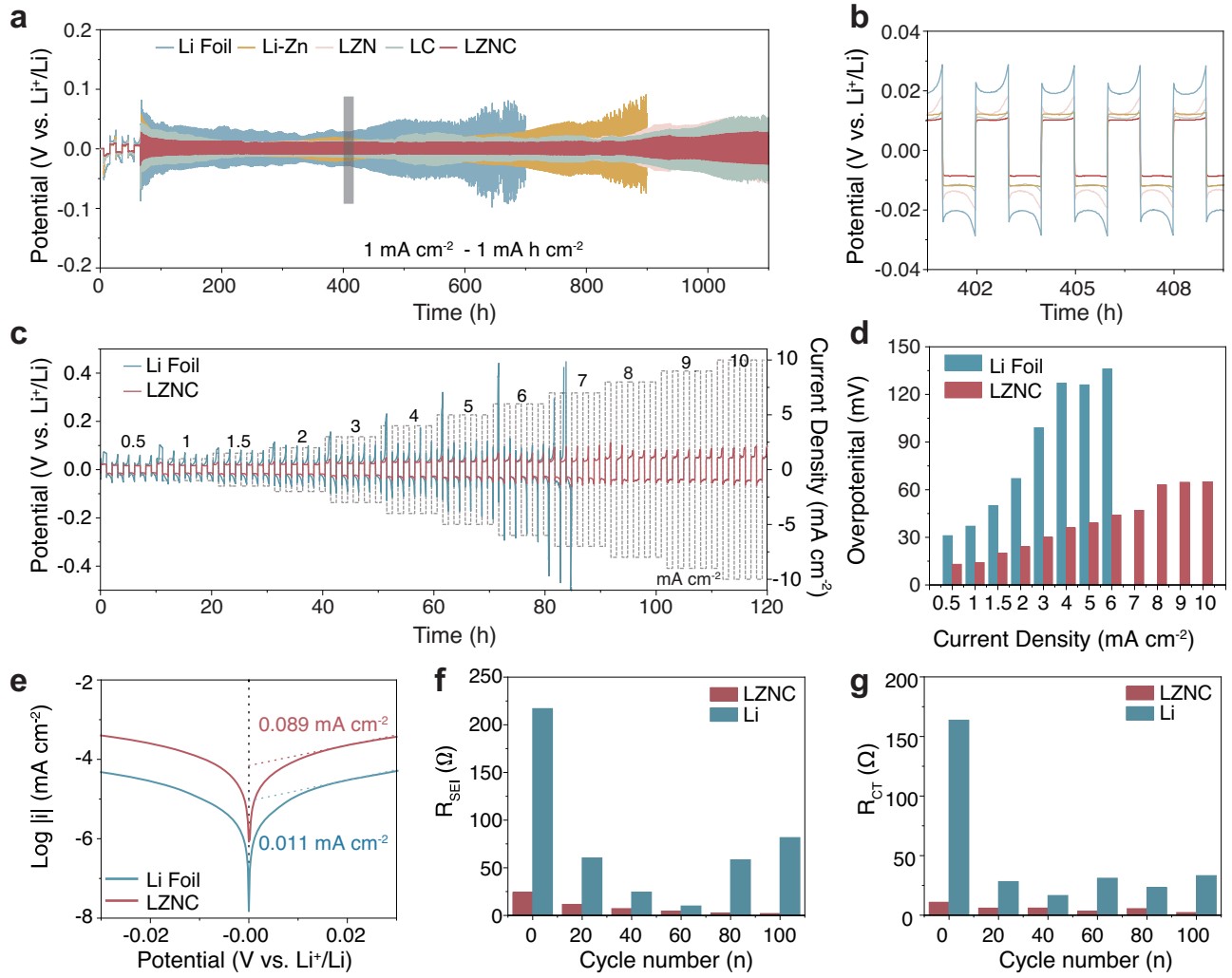

**Fig. 4 | Electrochemical performance of LZNC symmetric cells. a** Comparison of electrochemical Li plating/stripping performance and **b** the corresponding voltage profiles at selected times for bare Li, Li–Zn, LZN, LC, and LZNC symmetric cells at 1 mA cm$^{-2}$ and 1 mA h cm$^{-2}$. **c** Rate performances of bare Li and LZNC symmetric cells at the current densities from 0.5 to 10 mA cm$^{-2}$. **d** Comparison of the overpotentials of bare Li and LZNC symmetric cells at various current densities. **e** Tafel plots of bare Li and LZNC and symmetric cells. Bar charts comparing **f** R$_s$ and **g** R$_{ct}$ obtained from EIS spectrum fitting of LZNC and Li foil at different Li plating/stripping stages under 1 mA cm$^{-2}$ and 1 mA h cm$^{-2}$. The details are shown in Supplementary Table 4.

uniformly distributed within the CNTs matrix, without any signs of cracking or disintegration. This structural integrity was further confirmed by XRM (Fig. 3d, Supplementary Fig. 24), revealing that even after 100% Li stripping, the 3D Zn-CNTs network remains intact, accompanied by the formation of numerous pores within the delithiated scaffold. These interconnected voids accommodate subsequent lithium deposition, while CNTs reinforcement ensures mechanical coherence, rendering the LZNC negative electrode highly resistant to pulverization upon long-term cycling. We further investigated the structural and morphological evolution of lithium plating at 0.5 mA cm$^{-2}$ on a 25% delithiated bare Li and LZNC negative electrodes. The electrochemical profiles showed that both the nucleation overpotential and deposition overpotential of the LZNC negative electrode were markedly lower than those of bare Li, suggesting that the Li–Zn alloy, acting as lithophilic sites, promotes a and flat lithium deposition (Fig. 3g). SEM images reveal a rough and dendritic surface observed for the Li foil negative electrode, in sharp contrast with a planar and dendrite-free surface for the LZNC negative electrode (Supplementary Fig. 25). Upon increasing the current density to 2 mA cm$^{-2}$, the lithium deposition on the LZNC negative electrode maintains an orderly pattern (Supplementary Fig. 26).

The electrochemical performance of the LZNC foil negative electrode was first evaluated in symmetric cells. Figure 4a, b presents the voltage-time plots for bare Li, Li–Zn, LZN, Li-CNTs (LC), and LZNC symmetric cells, each with 50 μm thickness, at 1 mA cm$^{-2}$ with a capacity of 1 mA h cm$^{-2}$. The result shows that the LZNC foil negative electrode delivers a remarkably stable Li plating/stripping performance over 1200 h with a small voltage hysteresis of ~8 mV (Supplementary Fig. 27). By contrast, the bare Li and Li–Zn electrodes experience a quick short-circuiting, accompanied by hysteresis increase to around 100 mV within 600 and 900 h, respectively. As for LZN and LC symmetric cells, although the cycle life was extended compared to bare Li, their voltage hysteresis significantly increased upon Li plating/stripping (>50 mV), accompanied by non-smooth voltage profiles indicative of unstable interfacial kinetics. Even when the current density and areal capacity are increased to 5 mA cm$^{-2}$ and 5 mA h cm$^{-2}$, the LZNC symmetric cell can undergo stable Li plating/stripping for over 500 h (Supplementary Fig. 28). The rate performance of the symmetric cells was evaluated at various current densities ranging from 0.5 to 10 mA cm$^{-2}$ (Fig. 4c). As the current density increases, the bare Li cell exhibited significant voltage fluctuations, suggesting the formation of an unstable interface on the bare Li negative electrode. A sudden

voltage drop is observed at 7 mA cm$^{-2}$, which indicates the occurrence of a short circuit, which could be induced by the penetration of Li dendrites. In contrast, LZNC cells maintain stable Li plating/stripping with low voltage hysteresis (<62 mV at 10 mA cm$^{-2}$; Fig. 4d). The low overpotentials of the LZNC electrode indicate a reduced energy barrier for Li nucleation/plating, which facilitates a more homogenous and planar Li deposition. The favorable electrochemical performance under high current densities indicates the potential of LZNC negative electrodes applied under more practical conditions.

The kinetic properties of the LZNC foil electrode were further validated by the exchange current density ($i_0$) result derived from the Tafel plots (Fig. 4e, Supplementary Fig. 29). The $i_0$ of the LZNC foil electrode is 0.089 mA cm$^{-2}$, significantly higher than that of the bare Li electrode (0.011 mA cm$^{-2}$). Furthermore, as illustrated in Fig. S29, LZN foil exhibits a significant enhancement (0.0716 mA cm$^{-2}$) compared to the baseline LZ alloy (0.0131 mA cm$^{-2}$). In contrast, the LZC composite demonstrates only a marginal improvement (0.0161 mA cm$^{-2}$). The LZNC composite achieves the highest exchange current density (0.0887 mA cm$^{-2}$), which we attribute to the formation of an interwoven Li$_3$N-CNTs network that facilitates ion and electron transport. To elucidate the kinetics of the LZNC electrode, electrochemical impedance spectroscopy (EIS) measurements were conducted on symmetric cells at different Li plating/stripping stages under 1 mA cm$^{-2}$ and 1 mA h cm$^{-2}$. The high-frequency semicircle in the EIS data corresponds to the interfacial resistance ($R_s$), indicating lithium-ion diffusion within the solid electrolyte interphase (SEI) layer, while the low-frequency semicircle represents the charge transfer resistance ($R_{ct}$) at the electrode/electrolyte interface, reflecting the electrochemical interfacial kinetics. As shown in Fig. 4f, g, both $R_s$ and $R_{ct}$ of the LZNC electrode remain markedly lower than those of bare Li throughout Li plating/stripping, demonstrating a more stable and conductive interface. The total impedance of LZNC decreases initially and then stabilizes upon prolonged Li plating/stripping, suggesting progressive interfacial optimization and the formation of a robust conductive network (Supplementary Fig. 30). In contrast, the impedance of bare Li increases significantly upon Li plating/stripping, indicative of growing polarization and interfacial degradation. The activation energy ($E_a$) was calculated from the fitted profile of 1000/T and ln T/$R_s$ (Supplementary Fig. 31). The details of the references are shown in Supplementary Table 5, Supplementary Fig. 32. LZNC negative electrode presents a significantly lower $E_a$ (54.8 kJ mol$^{-1}$) compared to the bare Li (72.5 kJ mol$^{-1}$), indicating the fast Li-ion diffusion through the 3D interior interface of the LZNC foil. The enhanced kinetics can be ascribed to the interconnected and mechanically robust 3D conductive framework composed of LiZn, CNTs, and Li$_3$N, which facilitates rapid electron and ionic transport.

Post-mortem characterizations were then performed on the cycled electrodes to elucidate the electrochemical stability of the LZNC negative electrode. Top-sectional SEM images of the bare Li after Li plating/stripping at 1 mA cm$^{-2}$ and 1 mA h cm$^{-2}$ for 100 h revealed a porous surface structure with obvious cracks, suggesting inhomogeneous and dendritic lithium deposition (Fig. 5a). These cracks originated from the bare Li foil, which had poor mechanical strength and can hardly withstand significant strain during repeated lithium plating and stripping. This can lead to the disruption of the electronic pathway between the active material and the collector, ultimately causing irreversible capacity loss. In sharp contrast, the surface of LZNC after Li plating/stripping remains smooth and flat (Fig. 5b). From the cross-view SEM images (Fig. 5c), a huge crack was found between the fresh Li-deposition layer and the underneath unreacted Li electrode, which accounts for the large polarization and poor Li plating/stripping stability of bare Li. By contrast, LZNC foil preserved good structure integrity after Li plating/stripping without notable structural damage, indicating uniform and compact Li deposition (Fig. 5d). The structure of LZNC is stabilized by its pulverization-resistant characteristic, which

allows it to preserve the original 3D ionic and electronic conductive framework even after the metallic Li is completely stripped.

In-depth X-ray photoelectron spectroscopy (XPS) measurements were conducted on the negative electrodes after Li plating/stripping to investigate their interfacial evolution. As shown in Fig. 5e, Supplementary Figs. 33, 34, a strong signal of carbon-containing organic compounds derived from the decomposition of solvent was detected on the bare Li surface. In contrast, the peak signal intensity of the C 1s spectra on the LZNC surface was relatively weak (Supplementary Fig. 35). With increasing etching depth, the carbon signal decreased significantly, while the peak signal intensities of nitrogen (N) and zinc (Zn) remained almost constant (Fig. 5f–h). This indicates that the interface compositions on the surface and within the LZNC are dominated by inorganics. Furthermore, ex situ XPS at different Li plating/stripping stages (Supplementary Figs. 36, 37) reveals a dynamic interfacial evolution, verifying a gradual enrichment of inorganic species, mainly Li$_3$N, accompanied by a reduction of organic components. The appearance of a characteristic C−N peak after extended Li plating/stripping confirms the continuous enrichment of Li$_3$N on the CNTs surface, leading to the formation of a stable inorganic-rich SEI that ensures efficient Li$^+$ transport and mechanical robustness at the interface. AFM measurements (Supplementary Fig. 38) were further conducted to probe the mechanical evolution of the electrode interface during Li plating/stripping. The surface modulus of the LZNC electrode gradually increases with prolonged Li plating/stripping and remains consistently higher than that of bare lithium metal, reflecting a mechanically reinforced interface (Fig. 5i). This enhancement correlates well with the XPS results, which indicate the progressive enrichment of inorganic species within the interphase. The formation of this inorganic-rich layer effectively strengthens the mechanical integrity, suppresses lithium dendrite penetration, and preserves interfacial stability. Consequently, even after 300 cycles, the LZNC electrode maintains a smooth and compact surface morphology, indicative of uniform and dense lithium deposition (Supplementary Fig. 39). In contrast, bare Li electrodes after Li plating/stripping exhibit pronounced surface roughening, irregular dendritic growth, and significant accumulation of electrochemically isolated lithium. TOF-SIMS analyses (Fig. 5j–l) further confirm a homogeneous 3D distribution of Li, N, and Zn species within the LZNC electrode, validating the formation of a stable inorganic framework that ensures favorable ion transport and long-term dendrite suppression.

The potential application of the LZNC foil negative electrode was further evaluated in full cells by pairing it with the LiNi$_{0.8}$Co$_{0.1}$Mn$_{0.1}$O$_2$ (NCM811) positive electrode. The cells are evaluated at a current rate of 1 C (1 C = 200 mA g$^{-1}$) in a LiTFSI-containing commercial ester electrolyte with an N/P ratio of 4. In addition, a full cell using the same NCM811 positive electrode and 50-μm-thick bare Li negative electrode (Li|| NCM811) was also assembled for comparison. As shown in Fig. 6a, the LZNC||NCM811 cell exhibits a long lifespan over 500 cycles, with an average CE above 99.9% throughout the cycling. By contrast, the Li|| NCM811 exhibits rapid capacity decay after only 100 cycles (retention below 50%), with a lower CE < 80%. The low CE could be attributed to the rapid active-Li loss resulting from the formation of Li dendrites or dead Li (i.e., Li metal regions which are electronically disconnected from the current collector), therefore leading to a fast cyclic deterioration. From the discharge/charge voltage profile at different cycles displayed in Fig. 6b, Supplementary Fig. 40, the LZNC||NCM811 full cell exhibits lower potential hysteresis in both the charge and discharge processes compared with the full cell using a bare Li negative electrode, indicative of the better kinetics for LZNC. The high-rate capability of full cells was further evaluated. As shown in Fig. 6c, d, LZNC|| NCM811 cell delivers much higher discharge specific capacities at current rates ranging from 0.1 to 10 C (1.0 C = 200 mA g$^{-1}$), compared with Li||NCM811 cell. Even under a high rate of 10 C, the cell still delivers a high capacity of 117.7 mA h g$^{-1}$, which is much higher than

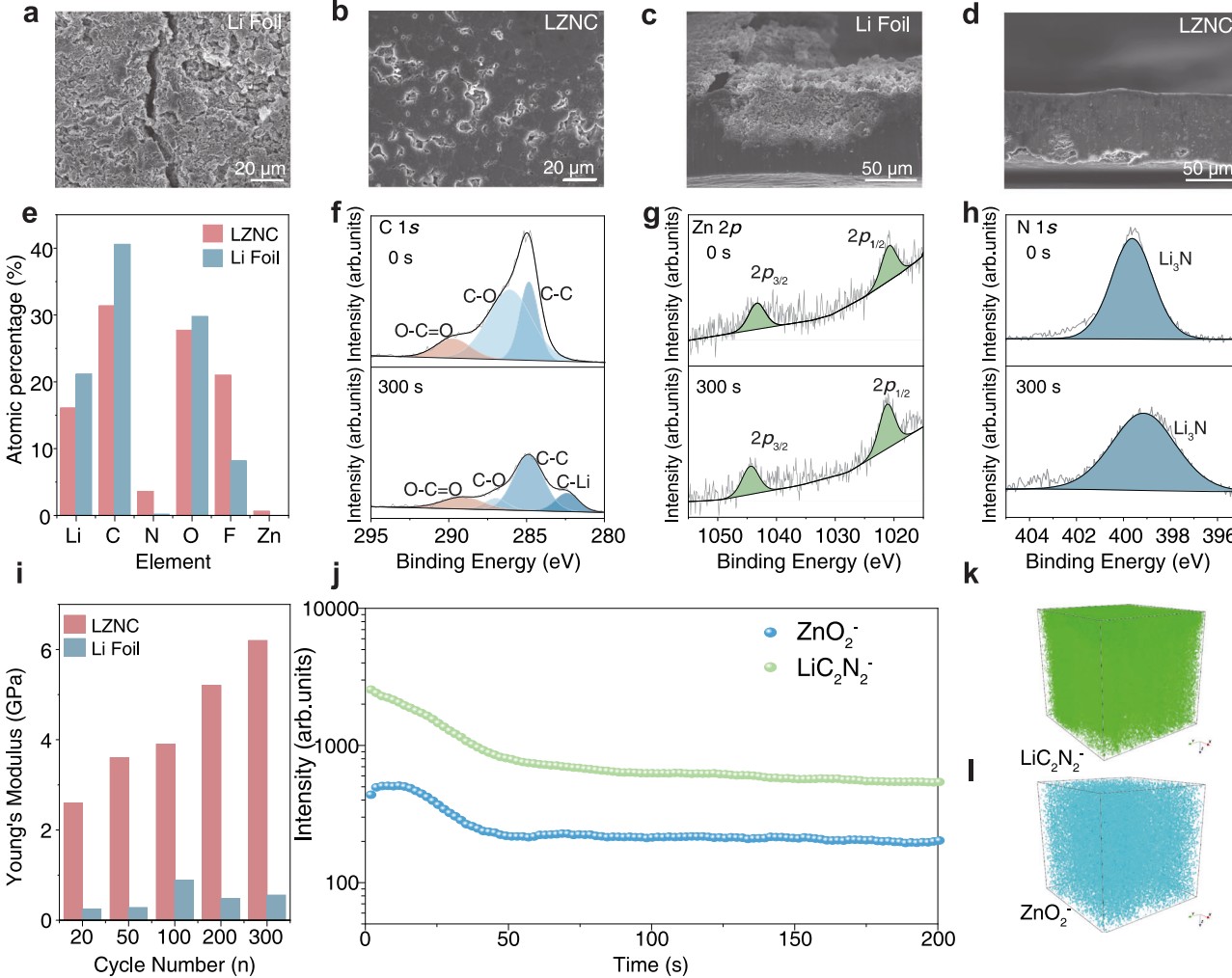

**Fig. 5 | Interfacial evolution and post-cycling characterization of the LZNC negative electrode. a**, **b** Top-view and **c**, **d** cross-sectional SEM images of bare Li and LZNC after 100 h at 1 mA cm$^{-2}$ /1 mA h cm$^{-2}$. **e** Comparison of atomic percentage on the surface of bare Li and LZNC negative electrodes from in-depth XPS measurements. XPS depth profile of **f** C 1$s$, **g** Zn 2$p$, and **h** N 1$s$ of LZNC negative electrode after 50 cycles of Li plating/stripping. **i** Comparison of AFM modulus values for LZNC and Li foil at different cycle stages. **j** TOF-SIMS analysis of the LZNC negative electrode after Li plated/stripped for 500 cycles and the spatial distributions of **k** LiC$_2$N$_2^-$ and **l** ZnO$_2^-$.

that of the Li||NCM811 cell (53.1 mA h g$^{-1}$). The high-rate performance of the LZNC||NCM811 cell can be attributed to the fast ion/electron transfer and enhanced kinetics within the LZNC negative electrode. Moreover, to demonstrate the broad applicability of the LZNC negative electrode, we extended its use beyond conventional lithium-containing positive electrodes to lithium-free systems such as Li−S batteries, which exhibit stable cycling over 200 cycles at 0.5 C (1.0 C = 1675 mA g$^{-1}$), highlighting its compatibility and interfacial stability across diverse battery chemistries (Supplementary Fig. 41).

More practical and harsh conditions were tested to verify the practicality and implementation of the LZNC foil negative electrode. The mass loading of the NCM811 positive electrode was further increased to 15 mg cm$^{-2}$ (areal capacity: 3 mA h cm$^{-2}$) to couple with LZNC and bare Li negative electrode. As shown in Supplementary Fig. 42, Li||NCM811 encounters a cliff-jumping capacity decay at 100 cycles (capacity retention below 60%), indicative of a rapid lithium depletion arising from inhomogeneous Li plating/stripping. By contrast, LZNC||NCM811 demonstrates a remarkebly prolonged cycle life over 180 cycles at 0.3 C, with a capacity retention above 90%. Even when the areal capacity increased up to 4 mA h cm$^{-2}$, LZNC||NCM811 still presents a stable cycling performance over 120 cycles at 0.3 C, with 90% capacity retention (Supplementary Fig. 43).

A laminated pouch cell was constructed by coupling a high-loading NCM811 positive electrode (8 mA h cm$^{-2}$) with a free-standing LZNC negative electrode (80 μm thickness) under stringent conditions, including a low N/P ratio (1.9) and minimal electrolyte usage (1.5 g Ah$^{-1}$). (Fig. 6e, Supplementary Fig. 44). Detailed cell parameters are summarized in Supplementary Table 6. The LZNC||NCM811 pouch cell delivers cycling stability with 91.7% capacity retention after 300 cycles at 0.5 C (Fig. 6f). The galvanostatic charge/discharge profiles demonstrate minimal voltage polarization with high reversibility (Supplementary Fig. 45). To further assess electrode stability under realistic operating conditions, in situ cell swelling measurements were performed on LZNC||NCM811 pouch cells (Supplementary Fig. 46). The cell exhibits markedly smaller thickness fluctuations than Li||NCM811, evidencing uniform Li stripping/plating and structural robustness of the LZNC framework. To systematically elucidate the practical advantages of LZNC negative electrodes, we conducted a comparative analysis of recent pouch cell studies, focusing on three critical metrics of cycling stability, cycle life, and normalized total negative electrode specific capacity (NSC). The NSC, defined as

$$NSC = \frac{Capacity_{negative\ electrode}}{(mass_{negative\ electrode} + mass_{current\ collector}) \times N/P}$$, quantifies the effective capacity contribution of the Li negative electrode to cell-level specific

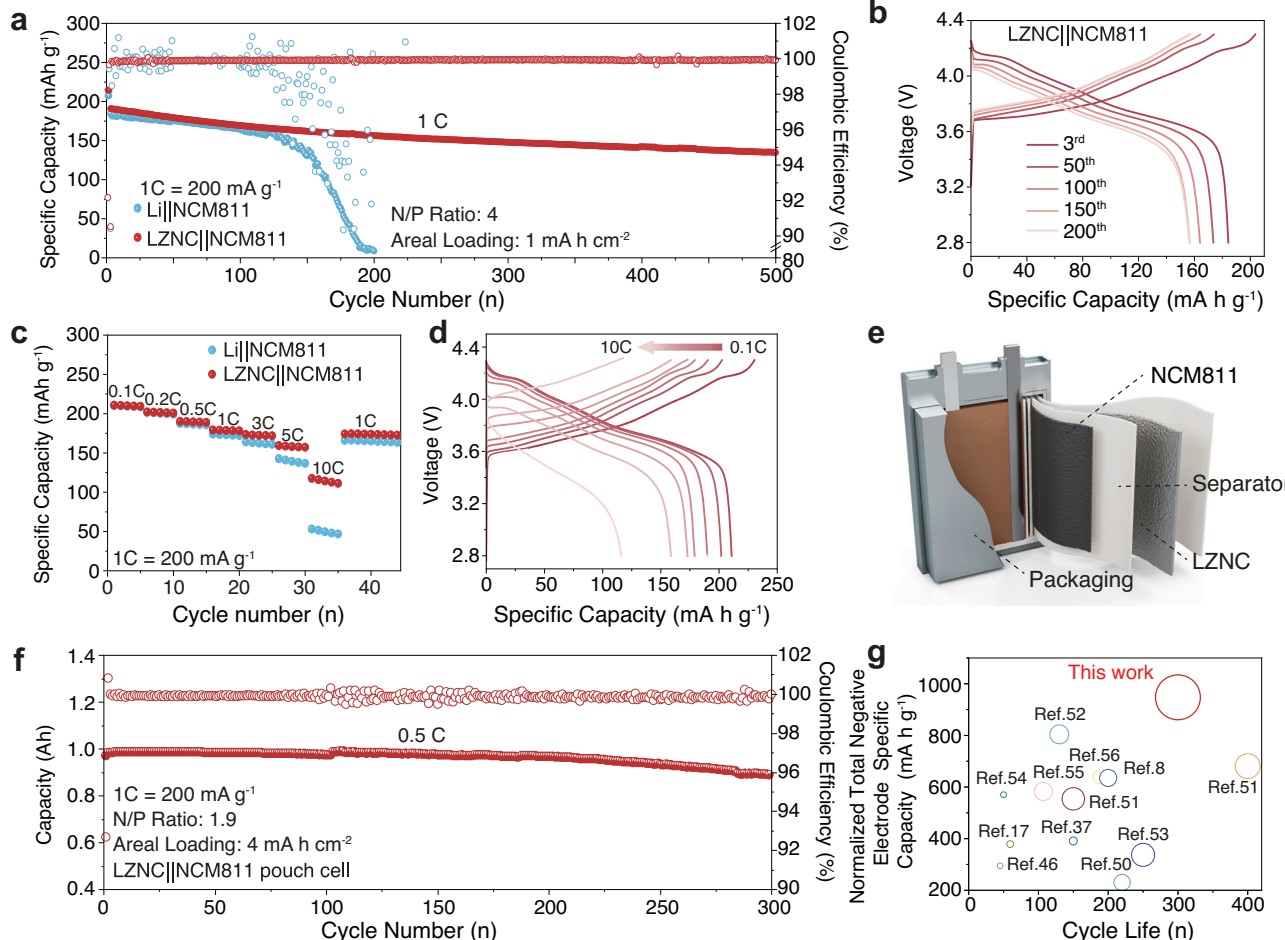

**Fig. 6 | Electrochemical performance of LZNC||NCM811 full cells (NCM811, 1 C = 200 mA g⁻¹). a** Cycling performance and **b** the corresponding voltage profiles of Li||NCM811 and LZNC||NCM811 coin-type full cells at 1C. **c** Rate performance of full cells at various rates from 0.1 C to 10 C. **d** Voltage profiles of LZNC||NCM811 full cells at different rates. **e** Schematic illustration of LZNC||NCM811 pouch cell.

**f** Cycling performance of LZNC||NCM811 pouch cells at 0.5 C. **g** Comparison of normalized total negative electrode specific capacity versus working cycle life in the previously reported Li-metal pouch cells. Note that the circumferential area inversely correlates with the capacity decay rate of the batteries [8,17,37,54,58–64]. Detailed parameters are provided in Supplementary Table 7.

energy by taking into account both the negative electrode/collector mass and the N/P ratio. Unlike conventional lithium-ion batteries, LMBs typically require an N/P > 1 configuration to offset irreversible lithium losses caused by their inherently low CE (< 99%). Though this strategy enhances cycling stability, it compromises specific energy due to excessive lithium loading. The NSC metric thus provides a more holistic evaluation of the role of the Li negative electrode in full-cell performance. As illustrated in Fig. 6g, the LZNC-based pouch cell achieves an NSC of 947 mA h g⁻¹, along with a prolonged cycle life. We also assembled 1 Ah LZNC |S pouch cells (employed a conventional DOL/DME-based electrolyte) with an areal capacity of 4.8 mA h cm⁻² (3 mg cm⁻²), which exhibit stable cycling over 60 cycles (Supplementary Fig. 41). To further validate its practical viability, an 8.5 Ah pouch cell was fabricated, demonstrating a gravimetric specific energy of 553 Wh kg⁻¹ at cell-level, including the weight of the pouch packaging (aluminum-plastic film and tabs). (Supplementary Fig. 47, Supplementary Table 8). The improvement in both specific-energy and cycle life verifies the effectiveness and potential of LZNC foil negative electrode for high-specific-energy, long-lifespan LMBs.

## Discussion

In conclusion, we have demonstrated a rationally designed, freestanding LZNC composite foil that transcends conventional negative electrode design limitations by simultaneously optimizing mechanical, electrochemical, and processing properties. The Li–Zn alloy strengthens tensile resistance and regulates lithium deposition, while the Li₃N-CNTs network reinforces mechanical toughness, collectively yielding a toughness of 1.3 × 10⁶ J/m³ (approximately a 1200% enhancement over bare Li). This mechanically tough framework facilitates the scalable production of thin foil negative electrodes (<10 μm), while preserving structural integrity during prolonged, deep-delithiated cycles. Simultaneously, the interior ion/electron transport network by 3D Li₃N-CNTs enhances charge transfer kinetics. Consequently, the LZNC||NCM811 full cell maintains stable electrochemical performance over 500 cycles and retains its rate capability up to 10 C. The Ah-grade pouch cell delivers 92% capacity retention after 300 cycles under the tested conditions, performing comparably to other recently reported Li-metal systems. Furthermore, an 8.5 Ah prototype pouch cell demonstrates a high practical specific energy of 553 Wh kg⁻¹ at the cell-level, including the weight of the pouch packaging (aluminum-plastic film and tabs). This study highlights the potential of rationally engineered 3D Li-composite foil negative electrodes in achieving a synergistic improvement in structure durability, manufacturability, high-rate capabilities, and long-term cyclability, offering a viable pathway toward high-specific-energy, high-performance LMBs.

## Methods

### Fabrication of Li–Zn, LC, LZN, and LZNC Composite Negative electrode

The Li–Zn (LZ) compound was synthesized by melting metallic lithium and zinc foils (Alfa, 99.98%) at a molar ratio of 20:1. The Li-CNTs (LC) composite was prepared by mixing metallic lithium with CNTs powder (XFM08, XFNANO) at a weight ratio of 4:1. The LZN compound was obtained by melting metallic lithium with $Zn_3N_2$ powders (Alfa, 99%) at a molar ratio of 60:1. All smelting reactions were carried out in an Ar-protected melting setup. To prepare LZC and LZNC, CNTs powder was added to the LZ and LZN slurries, respectively, at a mass ratio of 20%. All heating reactions described above were performed in an Ar-filled glovebox ($H_2O$ and $O_2$ < 0.1 ppm) using a melting furnace (KSL-1100X-SV-RM, Hefei Kejing Materials Technology, Ltd.) at 350 °C for 3 h, during which the melt was continuously stirred using a mechanical stirring paddle to ensure homogeneous mixing. The LZNC foil was directly employed as the negative electrode for cell assembly. Based on the precursor mass ratio (Li: $Zn_3N_2$: CNTs = 2: 1: 0.6), the expected reaction products include $Li_3N$, LiZn, excess Li, and CNTs. Assuming complete reaction between Li and $Zn_3N_2$, the masses of the products can be determined from their stoichiometry and molar masses. The resulting composite contains approximately 8.7 wt% $Li_3N$ and 16.7 wt% CNTs, giving a total $Li_3N$-CNTs fraction of ~25.4 wt% in the LZNC composite. After the reaction was completed, the melt was cooled to room temperature (25 °C). The solidified composite was then repeatedly rolled into foils using a rolling machine (MSK-HRP-1A, Hefei Kejing Materials Technology, Ltd.) in a dry room (relative humidity ~5%).

### Preparation of positive electrodes and fabrication of Li metal batteries

Electrochemical performances were evaluated in 2032-type coin cells. All the coin cells were fabricated in the Ar-filled glove box with water and oxygen concentration less than 0.1 ppm. Celgard 2500 polypropylene membranes were used as separators. All bare Li electrodes were prepared using 500 μm-thick Li foil. Sulfur (analytical reagent, Sigma-Aldrich) was combined with carbon nanotubes (CNTs, Nanjing XFNANO Materials) at a weight ratio of ~7:3. The mixture was transferred into a glass container, evacuated, and then heated at 155 °C for 20 h to achieve uniform dispersion. To prepare positive electrodes for Li–S coin cells, the sulfur-CNTs composites were blended with Super P conductive carbon and poly(vinylidene fluoride) (PVDF) binder at an 8:1:1 weight ratio, using N-methyl-2-pyrrolidone (NMP) as the solvent. The resulting slurry was uniformly applied onto carbon-coated aluminum foil and dried under vacuum at 60 °C for 24 h. Circular electrodes with a diameter of 10 mm were then punched from the coated foil for coin cell assembly. The procedure was adapted and modified from previously reported methods[45]. Two different areal capacities of $LiNi_{0.8}Co_{0.1}Mn_{0.1}O_2$ (NCM811) positive electrodes were prepared. Low areal capacity of NCM811 positive electrodes were used to probe the cycling stability of the negative electrodes. The NCM811 positive electrode slurry was prepared by mixing NCM811, Super P and PVDF in NMP with a mass ratio of 80:10:10. The areal loading of the NCM811 positive electrode was about 5.4 mg cm$^{-2}$. High areal capacity (3 and 4 mA h cm$^{-2}$) NCM811 positive electrodes were used to probe the application of negative electrodes in practical systems. NCM811, super P, and PVDF were mixed in a weight ratio of 96.8:1.9:1.9, and the active material loading was 15 and 20 mg cm$^{-2}$. The $LiNi_{0.98}Co_{0.01}Mn_{0.01}O_2$ positive electrode slurry was prepared by mixing $LiNi_{0.98}Co_{0.01}Mn_{0.01}O_2$, Super P and PVDF (polyvinylidene difluoride) in NMP (N-methyl-2-pyrrolidone) with a mass ratio of 98.1:1:0.9. The obtained slurries were cast on carbon-coated aluminum foil and dried at 80 °C for 12 h. The positive electrodes for coin cells were tailored into slices with a diameter of 10 mm. The thickness of the LZNC foil was adjusted to match the areal loading of the positive electrodes: for 1 mA h cm$^{-2}$

cathodes, a 20 μm-thick foil was used; for 3 mA h cm$^{-2}$ cathodes, a 100 μm-thick foil was employed; for 4 mA h cm$^{-2}$ cathodes, a 70 μm-thick foil was employed. For pouch cell assemblies, an 80 μm-thick foil was used, while symmetric cell tests were conducted with a 100 μm-thick foil. For liquid electrolyte cells, 80 μL ether-based electrolyte without any additives [1.0 M bis(trifluoromethane)sulfonimide lithium (LiTFSI, 99.0%, Sigma-Aldrich) dissolved in 1,3-Dioxolane (DOL, 99%, Alfa)/Dimethoxyethane (DME, 99%, Alfa) (1:1, by volume) containing 1% $LiNO_3$ (99.98%, Alfa)] was employed in symmetric cells and Li–S cells, and 80 μL commercial carbonate electrolytes [1.0 M lithium hexafluorophosphate ($LiPF_6$, 98%, Alfa) dissolved in ethylene carbonate (EC, 99%, Alfa)/ diethyl carbonate (DEC, 99%, Alfa) (1:1, by volume) containing 5% fluoroethylene carbonate (FEC, 98%, Alfa)] was applied for NCM811 full coin cells. LZNC∥NCM811 and LZNC∥$LiNi_{0.98}Co_{0.01}Mn_{0.01}O_2$ pouch cells were assembled with an 80-μm-thick LZNC composite foil and an LHCE [lithium bis(fluorosulfonyl)imide (LiFSI, 99.0%, Sigma-Aldrich), DME, and 1,1,2,2-tetrafluoroethyl-2,2,3,3-tetrafluoropropylether (TTE, Macklin, 99.5%) were mixed in a molar ratio of 1:1:3] electrolyte mass loading of 1.5 g A h$^{-1}$. The pouch cell was tested under a pressure of 0.2 MPa after being filled with electrolyte and sealed following the standard procedure. Additional details for the pouch cells can be found in Tables S6 and S8.

### Ex situ sample preparation and handling

After cycling, the cells were disassembled inside an Ar-filled glovebox ($H_2O$ and $O_2$ < 0.1 ppm). The metallic lithium or LZNC electrodes were carefully extracted, and residual lithium salts on the surface were removed by rinsing with DME. To maintain an inert atmosphere during subsequent characterization, the electrodes were transferred and mounted using dedicated protective holders or molds, ensuring that exposure to air or moisture was minimized. All manipulations were performed under controlled Ar conditions, and the average glovebox temperature was 25.0 °C.

### Characterizations

Particle size distribution is done by Nanoparticle Size and Zeta Potential Analyzer (Malvern, ZS90). The crystalline structure was analyzed by X-ray diffraction (Rigaku SmartLab) with Cu K$_\alpha$ radiation over the 2θ angles from 10° to 80° (step size: 0.02, scan speed: 2°/min). Tensile tests were conducted on a Universal Tensile Testing Machine (INSTRON 3365). Morphology characterization was carried out by Scanning electron microscopy (SEM, JEOL 8100) at 10 kV and transmission electron microscopy (TEM, JEM-2100F) with an accelerating voltage of 200 kV. 3D X-ray microscopy (ZEISS Xradia 510 Versa, Microscopy Customer Center Beijing) was utilized to obtain 3D images with a high resolution. In situ analysis of thickness expansion behavior was conducted using an In Situ Cell Swelling Analyzer (IEST, SWE2110). The measurements were performed on 1 Ah LZNC∥NCM811 and Li∥NCM811 pouch cells during 0.5 C charge-discharge cycling under a constant pressure of 0.2 MPa. Operando Optical Microscopy is performed using Leica optical microscopes. The adhesion of Li and microstructure of the Li surface after cycling were examined by using an atomic force microscope (AFM, Bruker DIMENSION ICON with a Nanoscope V controller) in an Ar-filled glove box with the AFM tip (Bruker Corp., k = 26 N m$^{-1}$, $f_0$ = 300 kHz) at a constant loading rate of 400 nm s$^{-1}$. The data of X-ray photoelectron spectroscopy (XPS) were collected through 150 W monochromatic Al K$_\alpha$ radiation ($h_v$ = 1486.6 eV) with different etch depths, and the base pressure was about $3 \times 10^{-9}$ mbar. The charge offset was applied by correcting the hydrocarbon C 1 s peak at 284.8 eV (Axis supra, Kratos Analytical Ltd.). Matrix-assisted laser desorption/ionization time-of-flight mass spectrometry (MALDI-ToF-MS) analyses were carried out using a Bruker UltrafleXtreme instrument. The composition and spatial distribution of ions on the anode surface, as well as in the depth-sputtered regions,

were examined via Time-of-Flight secondary ion mass spectrometry (ToF-SIMS, ToF-SIMS 5, ION-ToF GmbH, Münster, Germany). The ToF-SIMS system employed a 30 keV $Bi_3^+$ primary ion beam in combination with a 2 keV $Cs^+$ sputter beam, and an electron flood gun was applied for charge compensation. The experimental setup follows previously reported procedures[46]. The in situ X-ray diffraction patterns of all samples were collected using an X-ray diffractometer (D8 Bruker Advance) with Cu Kα radiation (λ = 1.5418 Å) in the scan range (2θ) of 10 - 100°. The samples for ex-SEM, ex-XPS, and AFM were obtained by disassembling the after-cycling batteries in an Ar-filled glove box without being exposed to air.

## Electrochemical measurements

For each electrochemical experiment, at least five cells were tested, and the data shown in the figures represent a representative cell. To evaluate Li plating/stripping performance, Li symmetric cells were tested under various areal current densities (1, 3, and 5 mA cm$^{-2}$) and areal capacities (1, 3 and 5 mA h cm$^{-2}$). In addition, the symmetric cells were first cycled under a low current density for 3 cycles to stabilize the SEI, with the cutoff capacity set the same as that used in the subsequent regular cycling. Charge-discharge tests were conducted with voltage ranging from 1.75–2.55 V for the Li∥S pouch cell (1 C = 1600 mA g$^{-1}$), 2.8-4.3 V for the Li∥NCM811 (1 C = 200 mA g$^{-1}$) coin cells, 2.8–4.4 V for the Li∥NCM811 pouch cell (1 C = 200 mA g$^{-1}$) and 2.5–4.5 V for the LZNC∥LiNi$_{0.98}$Co$_{0.01}$Mn$_{0.01}$O$_2$ pouch cell. Tafel plots were measured at a voltage range from −0.2 to 0.2 V (versus Li/Li$^+$) in Li symmetric cells with a scan rate of 1 mV s$^{-1}$ and then managed for linear fitting. The Electrochemical impedance spectroscopy (EIS) measurements were performed on a Princeton PARSTAT MC 1000 multi-channel electrochemical workstation over a frequency range of 0.1 Hz to 1 MHz with an amplitude of 10 mV at 25 °C. A total of 71 data points were collected. The measurements were carried out in potentiostatic mode after the cells were allowed to rest for 5 min at open-circuit voltage prior to the EIS scan. The activation energy ($E_a$) of Li$^+$ ions through SEI can be acquired via temperature-dependent EIS of the symmetrical cells from 293 to 333 K. Li$^+$ ion transportation is a thermally activated process, and it is suitable for the classical Arrhenius equation:

$$\frac{T}{R_{SEI}} = A \exp\left(-\frac{E_a}{RT}\right) \qquad (1)$$

where T is the absolute temperature, $R_S$ is the impendence of Li$^+$ across the SEI film, A is the pre-exponential constant, and R is the standard gas constant. According to the equivalent circuit, Rs was extracted at medium frequencies. ln(T/R$_s$) vs 1000/T can be well fitted by linear relationships. $E_a$ values can be obtained by $E_a$ = -R × slope (kJ mol$^{-1}$). Except for variable-temperature impedance testing, all electrochemical tests were conducted at room temperature (25 °C) inside a temperature-controlled test chamber.

## Computation detail

The ternary phase diagram of the Li−Zn-N system was generated via density functional theory (DFT) calculations based on thermodynamic analysis. It was performed via the pymatgen Python package, including the Materials project database[47]. It uses DFT as implemented in the Vienna Ab Initio Simulation Package (VASP) software[48] to evaluate properties of compounds. In Materials Project, A phase diagram module is a calculation of the thermodynamic phase equilibria of multicomponent systems[49–52]. Therefore, when using the Materials Project to construct phase diagrams, there is no need to provide data files containing the atomic coordinates of optimized computational models, as would be required for standard VASP calculations. This greatly accelerates the computation and application of material properties. Moreover, the workflow for phase diagram construction in this work is illustrated in Fig. S48 to facilitate a clearer understanding of the above description.

Molecular dynamics (MD) model system: The simulation box had dimensions of 34.36Å × 39.89Å × 38.48Å. Periodic boundary conditions were used in all three directions.

MD simulations: Molecular dynamics (MD) simulations were conducted using the LAMMPS package. Visualization of the simulation trajectories was performed with VMD software. The system was modeled using an all-atom CVFF force field, with a cut-off distance of 10 Å applied to all short-range van der Waals interactions. Long-range electrostatic interactions were calculated using the particle–particle particle–mesh (PPPM) method. For each system, five independent simulations with different initial velocity distributions were carried out, and all reported quantities represent averages over these runs. Simulations were performed in the canonical ensemble (NVT) at 648 K, with temperature controlled via the Nose–Hoover thermostat. The integration time step was set to 1 fs, and data were recorded every 1 ps. Each simulation was run for a total of 1 ns to ensure adequate sampling of the system behavior. The simulation protocol was adapted from standard MD procedures reported in the literature[46].

## Data availability

The authors declare that all data supporting the findings of this study are available within the paper and its supplementary information files. All raw data generated during the current study are available from the corresponding authors upon request. Source data are provided with this paper.

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

## Acknowledgments

Y.-H. Wang and S.-J. Tan contributed equally to this work. This work was supported by the National Key R&D Program of China (Grant No. 2021YFB2500301), Major Research plan of the National Natural Science Foundation of China (Grant No. 92372207), Strategic Priority Research Program of the Chinese Academy of Sciences (Grant No. XDB1040100), the CAS Project for Young Scientists in Basic Research (Grant No. YSBR-058), Basic Science Center Project of National Natural Science Foundation of China (Grant No. 52388201), the National Science Foundation of China (Grant No. 22379149 and 22309187), the Young Elite Scientists Sponsorship Program by CWAST (Grant No. 2022QNRC001), and the New Cornerstone Science Foundation through the XPLORER PRIZE. The authors thank to the support provided by Professor Zhao Yao in the field of TOF-SIMS.

## Author contributions

Y.-G. G.J.Z., and Y.-H.W. conceived the projects and designed the experiments. Y.-H.W. and S.-J.T. performed the experimental work with the help of C.-H.Z., J.-C.G., and X.-X.L. Y.-H.W., and S.-J.T. conducted the characterizations and data analysis with the help of R.-X.J., L.-B.H., X.-C.S., C.L., X.-S.Z., S.X., R.W., and X.Z. Y.-H.W. and J.Z. co-wrote the manuscript. All the authors discussed the results and commented on the manuscript.

## Competing interests

The authors declare no competing interests.
