## [Transparent Peer Review file · Nature Communications]

Engineering thin 3D Li-composite foil negative electrodes with high mechanical toughness

Corresponding Author: Professor Yu-Guo Guo

Version 0:

Reviewer comments:

Reviewer #1

(Remarks to the Author)

This work presents an innovative free-standing composite-foil anode featuring toughness reinforced 3D architecture through integration of Li-Zn alloy and Li₃N-enriched CNTs. The design constitutes a strategic resolution to the persistent trade-offs among mechanical robustness, manufacturing feasibility, and charge-transfer kinetics in lithium metal anodes. Critically, the exceptional fracture toughness of foil anode effectively mitigates structural degradation in ultrathin 3D configurations during deep-delithiation cycles. The demonstration over 300 stable cycles, coupled with a high energy density of 553 Wh kg⁻¹ in Ah-scale pouch cells, provides compelling evidence of its practical viability. This study presents foundational design principles for next-generation ultrathin 3D composite lithium anodes, making a significant contribution to the field of lithium metal batteries. Due to its rigorous investigation and innovative approach, I believe this work will generate broad interest within both academic and industrial communities. Therefore, I recommend its publication in Nature Communications. To further enhance the clarity and impact of the manuscript, I suggest addressing the following revisions.

1. In Figure 4a, what is the overpotential of the LZNC electrode at 1 mA cm⁻²?
2. Was the rate test in Figure 6c conducted under temperature-controlled conditions? If so, please specify the temperature, as it could influence high-rate performance.
3. What is the mass ratio of the Li₃N-CNTs in the LZNC composite? This information would help clarify their role in reinforcing the mechanical property of the ternary composite.
4. Which component or structural feature of the LZNC architecture primarily contributes to the enhancement in toughness? Additionally, in Figure 4e, could the authors elaborate on the physical significance of the exchange current density?
5. What is the theoretical gravimetric specific capacity of the LZNC composite anode, and does it match with the tested value?
6. Others: When drawing EIS diagrams (such as Figure 4f and S23-24), it is best to keep the length of the x-axis and y-axis consistent, that is, square; There are missing spaces between some units and numbers, such as in the Computation detail section and Figure 3f-h; Some abbreviations are not given their full names when they first appear, such as SEI; In the vertical axis units of images 3e, 6b, 6d, and S32-33, there is an additional symbol after V;

Reviewer #2

(Remarks to the Author)

The manuscript presents the engineering of a three-dimensional Li-composite foil anode (LZNC), combining Li-Zn alloy with Li₃N-enriched carbon nanotubes (CNTs). The authors report enhanced mechanical toughness, excellent tensile strength, and high rupture toughness enabling ultra-thin anodes. The LZNC anode demonstrates superior electrochemical stability, minimal volume fluctuation during cycling. However, I believe this manuscript is not yet suitable for publication in its current form. Below are my constructive comments:

1. It is still required to clarify the precise roles of Li₃N and CNTs in enhancing mechanical properties and conductivity, supported by explicit experimental evidence.
2. There is limited rigorous benchmarking against current state-of-the-art 3D composite anodes, essential for substantiating the claimed advantages.
3. It would be better to provide a deeper analysis of long-term electrochemical stability, specifically detailing the mechanism of Li deposition and interface evolution with quantitative in-depth characterization.

4. The electrochemical and structural characterizations, while extensive, still lack critical in-situ insights into the structural dynamics during actual battery operation conditions.
5. Post-mortem characterization of the cycled electrodes is necessary to support claims regarding durability and interfacial stability.

Reviewer #3

(Remarks to the Author)

The three-dimensional lithium metal composite anode (LZNC) proposed in this study aims to resolve the contradictions between mechanical toughness, processability, structural integrity, and electrochemical performance of ultrathin lithium metal anodes by synergistically integrating Li-Zn alloy with Li₃N-enriched carbon nanotubes (Li₃N-CNTs). However, there are some unclear points in this manuscript, with a few flaws in the experimental data. Furthermore, the innovative aspects of this research have not been fully explored. Based on these considerations, major revisions are required before its publication in Nature Communications.

1. As can be seen from Figure 1a, the authors added CNTs during the smelting process. Then, at high temperatures, why did CNTs not react with Li to form Li-Cx? If Li-Cx was formed, would it change the fibrous structure of CNTs? If the fibrous structure of CNTs were not changed, more intuitive morphological structure characterization should be provided.
2. As mentioned by the authors in Figure 3a, "the voltage gradually rises while the peak intensity of LiZn diminishes, accompanied by the emergence of the peak of Zn". It is crucial to clarify whether this dealloying process is reversible, which is highly relevant to the structural integrity of the 3D framework during lithium deposition/stripping with large areal capacities.
3. According to Figure 6f and Figure S33, it can be concluded that the assembled pouch cell has a capacity of 7.6 Ah and a mass of 0.058 kg. The median voltage of NCM811 is generally 3.8 V, and based on this calculation, the overall energy density is only 497.9 Wh/kg. However, the data calculated in the manuscript is 553 Wh/kg. A reasonable explanation should be provided.
4. The LZNC lithium-containing composite prepared in this manuscript exhibits good cycling performance when matched with the lithium-containing NCM811 cathode. However, it remains unclear whether it has been matched with other cathodes, such as fabricating Li||S pouch cells to test its cycling performance with lithium-free cathodes.
5. In Figure 1c, the scale bars for the 8 μm thickness test are misaligned. Please carefully check and correct this.
6. In Figure S22, for the LZNC symmetric cell, the polarization voltage increases significantly at around 150 h of cycling and then decreases slowly. It is necessary to clarify the underlying mechanism for this phenomenon.
7. In Figure 4c, it is suggested that different current densities should be distinguished, and the corresponding current density values should be indicated.
8. Concerning Figures 6a and 6f, it is recommended that the corresponding N/P ratios and cathode loadings be labeled in the figures to ensure data completeness.
9. Figure 2a presents the XRD data of the LZNC composite, but there is an absence of peaks corresponding to lithiated carbon nanotubes. A reasonable explanation for this should be provided.

Version 1:

Reviewer comments:

Reviewer #1

(Remarks to the Author)

The manuscript has been thoroughly revised in accordance with all comments, and I believe it now meets all requirements for publication.

Reviewer #2

(Remarks to the Author)

The manuscript has been revised accordingly. There is no more comment.

Reviewer #3

(Remarks to the Author)

The authors have addressed the concerns, and it can be published at this stage.

Detailed lists of our point-by-point responses to the comments from the three reviewers are listed below.

For Reviewer #1

General Comments: This work presents an innovative free-standing composite-foil anode featuring toughness reinforced 3D architecture through integration of Li-Zn alloy and Li₃N-enriched CNTs. The design constitutes a strategic resolution to the persistent trade-offs among mechanical robustness, manufacturing feasibility, and charge-transfer kinetics in lithium metal anodes. Critically, the exceptional fracture toughness of foil anode effectively mitigates structural degradation in ultrathin 3D configurations during deep-delithiation cycles. The demonstration over 300 stable cycles, coupled with a high energy density of 553 W h kg⁻¹ in Ah-scale pouch cells, provides compelling evidence of its practical viability. This study presents foundational design principles for next-generation ultrathin 3D composite lithium anodes, making a significant contribution to the field of lithium metal batteries. Due to its rigorous investigation and innovative approach, I believe this work will generate broad interest within both academic and industrial communities. Therefore, I recommend its publication in Nature Communications. To further enhance the clarity and impact of the manuscript, I suggest addressing the following revisions.

Response to General Comments: We sincerely thank the reviewer for the encouraging evaluation of our work and for the thoughtful suggestions. We are pleased that the reviewer recognizes the novelty, practicality, and significance of our LZNC composite anode design. To further improve the clarity and quality of the manuscript, we have carefully addressed all the comments and revised the text and figures accordingly. Please find our point-by-point responses below.

Comment (1): In Figure 4a, what is the overpotential of the LZNC electrode at 1 mA cm⁻².

Responses to Comment 1: We appreciate the reviewer's attention to the overpotential, as it is a key parameter reflecting the interfacial kinetics and electrochemical stability of the electrode. As shown in Figure R1, under the condition of 1 mA cm⁻²-1 mA h cm⁻²

², the LZNC anode exhibits an overpotential of ~ 8 mV during the stable cycling stage, reflecting the highly stable electrode/electrolyte interface. In the revised manuscript, we have included more discussion in Supplementary materials to clarify the point (*Text of Fig. S27*) and updated the manuscript accordingly (*Page 9-10*).

Fig. R1. Voltage profiles of LZNC symmetric cell under 1 mA cm^{-2} - 1 mA h cm^{-2} conditions

Comment (2): Was the rate test in Figure 6c conducted under temperature-controlled conditions? If so, please specify the temperature, as it could influence high-rate performance.

Responses to Comment 2: We appreciate the reviewer’s thoughtful question regarding the testing conditions, as temperature can significantly influence the rate performance and interfacial behavior of lithium metal anodes. We confirm that all rate performance measurements shown in Fig. 6c were conducted at room temperature (approximately 25°C), without any external heating or cooling applied. This further validates the practicality of our design under standard operating conditions.

In the revised manuscript, we have added further explanations to the manuscript (*Page 19*).

Comment (3): What is the mass ratio of the Li_3N -CNTs in the LZNC composite? This information would help clarify their role in reinforcing the mechanical property of the ternary composite.

Responses to Comment 3: Thank you for this important question. Based on the precursor mass ratio ($\text{Li} : \text{Zn}_3\text{N}_2 : \text{CNTs} = 2 : 1 : 0.6$), the expected reaction products are Li_3N , LiZn , excess Li , and CNTs . Assuming complete reaction between Li and Zn_3N_2 ,

the mass of each product can be determined from their stoichiometry and molar masses. The resulting composite mass fractions are approximately 8.7% for Li_3N and 16.7% for CNTs, yielding a total Li_3N -CNTs content of about 25.4% in the LZNC composite. These components together form a mechanically robust network that significantly reinforces the toughness and structural integrity of LZNC.

In the revised manuscript, we have included more discussion in Supplementary materials to clarify the point (*Text of Table. S2*) and updated the manuscript accordingly (*Page 17*).

Table R1. Mass fractions of the components in the LZNC composite.

Component	Mass Fraction (%)	Role
LiZn	26.9%	Deposition site
		Second phase reinforcement
Li_3N	8.7%	Ionic transfer highway
		Electronic transfer highway
CNTs	16.7%	3D interwoven matrix
Excess Li	47.7%	Electrochemically active material

Comment (4): Which component or structural feature of the LZNC architecture primarily contributes to the enhancement in toughness? Additionally, in Figure 4e, could the authors elaborate on the physical significance of the exchange current density?

Responses to Comment 4: Thank you for this insightful question. The remarkable enhancement in toughness of the LZNC composite primarily arises from the synergistic effect between the Li-Zn alloy and the Li_3N -CNTs network. To better demonstrate the contribution of each component to the mechanical properties of LZNC, we further conducted complementary tensile tests on these foils to evaluate their mechanical properties. As shown in Figs. R2 and R3, the Li-Zn alloy provides high tensile strength and resistance to fracture, while the Li_3N -CNTs network contributes mechanical toughness and structural flexibility. Together, these components form a robust 3D framework that effectively resists crack propagation and pulverization during cycling.

Regarding the exchange current density shown in Figure 4e, it reflects the intrinsic

interfacial reaction kinetics between the electrode and electrolyte. Physically, it defines the rate of lithium redox reactions at equilibrium, and thus governs how rapidly the system can respond to applied current. A higher exchange current density corresponds to faster interfacial charge-transfer kinetics, enabling lower overpotentials under practical current densities. Therefore, the elevated exchange current density observed in the LZNC anode highlights its favorable reaction kinetics.

In the revised manuscript, we have updated figures (*Text of Figs. 1b and S8*) and included more discussion (*Page 4*).

Fig. R2. Tensile stress-displacement profiles of Li, Li-Zn, LZN, LZC and the LZNC foil, with the integrated area beneath the stress-strain curve (shaded region) quantifying its fracture toughness.

Fig. R3. Columnar Comparison Chart of Tensile Strength and Toughness for Li, LZ, LZN, LZC, and LZNC.

LZN, LZC, and LZNC.

Comment (5): What is the theoretical gravimetric specific capacity of the LZNC composite anode, and does it match with the tested value?

Responses to Comment 5: Thank you for raising this important question. Based on the precursor ratio and assuming complete reaction, the major components in the LZNC composite are LiZn (26.9 wt%), Li₃N (8.7 wt%), CNTs (16.7 wt%), and excess metallic Li (47.7 wt%). Among these, the excess Li is electrochemically active and contributes to the reversible capacity. Given the theoretical specific capacity of lithium (3860 mAh g⁻¹), the overall theoretical gravimetric specific capacity of the LZNC composite can be calculated as:

$$\text{Capacity} = 47.7\% \times 3860 \text{ mA h g}^{-1} = 1842 \text{ mA h g}^{-1}$$

This value matches well with the experimental results shown in Fig. 1e, where a reversible capacity of approximately 1800 mA h g⁻¹ is achieved at a current density of 0.5 mA cm⁻². The close agreement between theoretical and experimental values confirms the high utilization of the Li within the LZNC composite.

In the revised manuscript, we have included more discussion in Supplementary materials to clarify the point (*Text of Table. S2*) and updated the manuscript accordingly (*Page 4*).

Comment (6): Others: When drawing EIS diagrams (such as Figure 4f and S23-24), it is best to keep the length of the x-axis and y-axis consistent, that is, square; There are missing spaces between some units and numbers, such as in the Computation detail section and Figure 3f-h; Some abbreviations are not given their full names when they first appear, such as SEI; In the vertical axis units of images 3e, 6b, 6d, and S32-33, there is an additional symbol after V;

Responses to Comment 5: We sincerely thank the reviewer for the careful checking of figures and formatting details. Following the suggestion, we have revised the Electrochemical Impedance Spectroscopy (EIS) diagrams (Figs. 4f and S23-24) to keep the x- and y-axis lengths consistent (square format). We have also carefully checked the manuscript and corrected all spacing issues between numbers and units in the

Computation Details section and figure captions. In addition, we have ensured that all abbreviations are given in full at their first appearance.

For the vertical axis labels in Figures 3e, 6b, 6d, and S32-33, the redundant symbols after “V” have been removed. With regard to the reviewer’s comment on “Figure 3f-h,” we could not locate these subfigures in our manuscript and believe the reviewer may be referring to Figures 5f-h. We have carefully re-examined Figures 5f-h and corrected the unit formatting issues accordingly.

To make the corrections clear, the revised figures are provided in the response letter as Figs. R4-R8. We greatly appreciate the reviewer’s attention to detail, which has helped us significantly improve the overall quality and readability of the manuscript. We have carefully checked all figures to avoid similar issues in the revised version. In the revised manuscript, we have made the corresponding modifications.

Fig. R4. Nyquist plots of bare Li and LZNC symmetric cells after 20 cycles.

Fig. R5. Nyquist plots of bare Li and LZNC symmetric cells before cycling.

Fig. R6. Nyquist plots of symmetric cells with Li foil and LZNC, respectively at a) 293K, b) 303K, c) 313K, d) 323K, e) 333K.

Fig. R7. XPS depth profile of f) C 1s, g) Zn 2p, and h) N 1s of LZNC anode after cycled for 50 cycles.

Fig. R8. a) Voltage profiles of 25% Li-stripped bare Li and LZNC anodes with Li plating at $0.5 \text{ mA cm}^{-2}/6 \text{ mA h cm}^{-2}$. b) the corresponding voltage profiles of NCM811||Li and NCM811||LZNC coin-type full cells at 1C. c) Voltage profiles of NCM811||LZNC full cells at different rates. d) The corresponding voltage profiles of NCM811||Li full cells. e) Voltage profiles of the NCM811||LZNC pouch cell. f) Voltage profiles of the 8.5 Ah LiNi_{0.98}Co_{0.01}Mn_{0.01}O₂||LZNC pouch cell.

For Reviewer #2

General Comments: The manuscript presents the engineering of a three-dimensional Li-composite foil anode (LZNC), combining Li-Zn alloy with Li₃N-enriched carbon nanotubes (CNTs). The authors report enhanced mechanical toughness, excellent tensile strength, and high rupture toughness enabling ultra-thin anodes. The LZNC anode demonstrates superior electrochemical stability, minimal volume fluctuation during cycling. However, I believe this manuscript is not yet suitable for publication in its current form. Below are my constructive comments:

Response to General Comments: Thank you for your positive comments on our manuscript. We have carefully considered each of the reviewer's constructive suggestions, which have helped us significantly improve the clarity and depth of our manuscript. In the revised version, we have addressed all the raised concerns point-by-point, clarified key technical aspects, and revised the relevant sections accordingly.

Comment (1): It is still required to clarify the precise roles of Li₃N and CNTs in enhancing mechanical properties and conductivity, supported by explicit experimental evidence.

Responses to Comment 1:

We thank the reviewer for this insightful comment. To unequivocally decouple the respective contributions of Li₃N and CNTs to the performance of the LZNC composite, we systematically prepared and characterized a series of foil anodes with controlled compositions, which are bare Li, Li-Zn (LZ), Li-Zn-CNTs (LZC), Li-Zn- Li₃N (LZN), and Li-Zn- Li₃N -CNTs (LZNC) composites.

We first employed Tafel measurements on symmetrical cells assembled from these foil anodes to probe their electrochemical reaction kinetics. The exchange current density, derived from Tafel measurements (Fig. R9), shows that the LZN foil exhibits a significant enhancement (0.0716 mA cm⁻²) compared to the baseline LZ alloy (0.0131 mA cm⁻²). In contrast, the LZC composite demonstrates only a marginal improvement (0.0161 mA cm⁻²). The LZNC composite achieves the highest exchange current density (0.0887 mA cm⁻²), which we attribute to the formation of an interwoven Li₃N-CNTs

network that facilitates fast ion and electron transport. This result highlights that the Li_3N phase is the primary contributor to the superior ionic conductivity, consistent with its known high intrinsic Li^+ ion conductivity and its role in establishing percolating ion-conduction pathways.

We further conducted complementary tensile tests on these foils to evaluate their mechanical properties (Figs. R10, R11). Quantitatively, the LZN composite exhibits a $\sim 200\%$ increase in tensile strength over the LZ baseline, an effect we ascribe to the second-phase reinforcement by the inorganic Li_3N particles. Notably, the LZC composite shows an even greater strength enhancement of $\sim 320\%$. Furthermore, we calculated the toughness, defined as the energy absorbed prior to fracture (the integral under the stress-strain curve, shown in Fig. R11), for each material. While LZN ($6.4 \times 10^5 \text{ J/m}^3$) shows a $\sim 100\%$ increase over LZ ($3.2 \times 10^5 \text{ J/m}^3$), LZC achieves a remarkable $\sim 240\%$ enhancement ($11 \times 10^5 \text{ J/m}^3$). The synergistic combination of both components in the LZNC composite yields the highest toughness value ($13 \times 10^5 \text{ J/m}^3$), representing a 300% enhancement relative to LZ. These data demonstrate that the CNTs network serves as the principal mechanical reinforcement, drastically improving both tensile strength and toughness by acting as a resilient scaffold that bridges Li-Zn domains, inhibits crack propagation, and effectively redistributes mechanical stress.

In summary, these complementary experimental analyses clearly delineate the distinct roles of each component wherein Li_3N is paramount for enhancing electrochemical kinetics, while CNTs are dominant in imparting mechanical resilience. Critically, our one-step melt-infusion strategy to obtain LZNC achieves intimate interfacial coupling between Li_3N and CNTs. This integration of LZNC facilitates a synergistic effect, enabling simultaneous and balanced improvements in ionic/electronic transport and structural robustness that both higher than that of either component alone. These findings elucidate the exclusive role of Li_3N and CNTs on the LZNC composite and contribute for the superior performance of our LZNC composite anode.

In the revised manuscript, we have updated figures (*Text of Figs. 1b, 4e, S8 and*

S29) and included more discussion (Page 4 and 10).

Fig. R9. Tafel plots of bare Li, Li-Zn, LZN, LZC and LZNC and symmetric cells.

Fig. R10. Tensile stress-displacement profiles of Li, Li-Zn, LZN, LZC and the LZNC foil, with the integrated area beneath the stress-strain curve (shaded region) quantifying its fracture toughness.

Fig. R11. Columnar Comparison Chart of Tensile Strength and Toughness for Li, LZ,

LZN, LZC, and LZNC.

Comment (2): There is limited rigorous benchmarking against current state-of-the-art 3D composite anodes, essential for substantiating the claimed advantages.

Responses to Comment 2: We appreciate the reviewer's insightful suggestion. To rigorously benchmark the advantages of our LZNC composite anode, we have conducted a comprehensive comparative analysis against current state-of-the-art 3D composite anodes, evaluating both the *intrinsic material properties* and the *resulting full-cell performance*.

From a materials perspective, Figure R12 and Table R2 provide a systematic comparison of gravimetric capacity, volumetric capacity, and electrode thickness for state-of-the-art alloy-based and carbon-based 3D Li-composite anodes. As summarized, lightweight carbon-based 3D composites typically achieve high gravimetric capacity, however, their low tap density and the inherent brittleness of many carbon frameworks often limit their volumetric capacity and processability, presenting a significant barrier to fabricating ultra-thin, Li-excess anodes for high-energy-density batteries. On the other hand, alloy skeletons facilitate high volumetric utilization but invariably sacrifice gravimetric capacity due to their heavier mass. Furthermore, these alloys often suffer from structural instability during cycling due to their discontinuous frameworks. In contrast, our LZNC architecture synergistically integrates a Li-Zn alloy with a Li₃N-enriched CNTs network. This unique design simultaneously achieves both high gravimetric (1800 mA h g⁻¹) and volumetric capacity (1915 mA h cm⁻³), overcoming the traditional trade-off between these metrics. Critically, as shown in Figure R10, the LZNC composite exhibits the highest tensile strength (23.9 MPa) and toughness (13 × 10⁵J/m³). This pronounced mechanical robustness enables the fabrication of ultra-thin lithium composite foils suitable for high-energy-density systems.

Fig. R12. Comparison of volumetric/gravimetric specific capacity of the previously reported 3D Li-C and 3D Li-alloy anodes. Note that the circumferential area inversely correlates with the thickness of the composite anodes.^[1-14] Detailed parameters are provided in Supplementary Table 2.

Table R2. Comparison of volume and gravimetric specific capacity for reported composite anode.

	Composite material	Gravimetric specific capacity (mA h g ⁻¹)	Volume specific capacity (mA h cm ⁻³)	Thickness (μm)	Ref.
Li-C	Li/3D hollow carbon fiber	1107	363	165	Joule 1 , 563–575 (2017)
	Li/hierarchical silver-nanowire-Graphene	1403	342	350	Adv. Mater. 30 , 1804165 (2018)
	Li/CNTs-MC	1640	793	126	Adv. Mater. 31 , 1805654 (2019)
	Li/coralloid carbon fiber	1570	666	150	Joule 2 , 764–777 (2018)
	Li@GDD-CH	1598	652	46	Angew. Chem. Int. Ed. 63 , e202403399 (2024)
	ZOS-CF@Li	1115	584	137	Adv. Funct. Mater. 35 , 2420382 (2025)
Li-Alloy	Li/Li ₂₂ Sn ₅ /LiF	606	1599	104	Adv. Mater. 35 , 2207310 (2023)
	Al-HCGB-Li	968	1776	50	Sci. Adv. 8 , eabq3445 (2022)
	Li/Li ₂₂ Sn ₅	656	841	400	Nat. Commun. 11 , 829 (2020)
	LiMg/CuCM	1401	633	104	Adv. Mater. 34 , 2205677 (2022)
	Li-B@SSM	996	1445	98.35	Adv. Mater. 35 , 2211203 (2023)
	Fe-N@SSM-Li	628	571	105	Adv. Funct. Mater. 33 , 2308022 (2023)
	3D Li/Li ₂₂ Sn ₅	1178	1798	100	Adv. Energy Mater. 13 , 2302755 (2023)
	TFA	1078	1552	100	Adv. Mater. 37 , 2506298 (2025)
This work	1799	1915	8		

From the full-cell performance perspective, to evaluate its performance under practical cell-level conditions, we benchmarked our pouch cells against current state-of-the-art systems (Figure R13 and Table R3). To facilitate a fair and comprehensive

assessment, we introduced a new evaluation parameter, the normalized anode-specific capacity (NSC). The NSC, defined as $NSC = \frac{Capacity_{anode}}{(mass_{anode} + mass_{current\ collector}) \times N/P}$, incorporates the influence of both the current collector and the N/P ratio, thereby enabling a direct and equitable comparison of the anode's effective capacity contribution within full-cell configurations. Unlike conventional lithium-ion batteries, LMBs typically require an N/P > 1 configuration to offset irreversible lithium losses caused by their inherently low Coulombic efficiency (<99%). Though this strategy enhances cycling stability, it compromises energy density due to excessive lithium loading. The NSC metric thus provides a more holistic evaluation of the role of Li anode in full-cell performance. As illustrated in Fig. 6g, the LZNC-based pouch cell outperforms previously reported systems, achieving a record-high NSC of 947 mA h g⁻¹, as well as prolonged cycle life and an outstanding per-cycle capacity retention of 99.94%. In contrast to prior studies that frequently emphasize either high (anode) capacity or long cycle life in isolation, the LZNC anode delivers both simultaneously under realistic pouch-cell conditions, benchmarking favorably against the most advanced Li-metal systems reported.

In the revised manuscript, we have included more discussion in the text and Supplementary materials to clarify the point (*Text of Figs. 1f and 6g, Tables S3 and S6*).

Fig. R13. Comparison of normalized total anode specific capacity versus working cycle life in the previously reported Li-metal pouch cells. Note that the circumferential area

inversely correlates with the capacity decay rate of the batteries.^[15-22] Detailed parameters are provided in Supplementary Table 5.

Table R3. Comparison of cycling performance for previously reported pouch cells.

(Total specific capacity of anode incorporates the influence of both the current collector and the N/P ratio, thereby enabling a direct and equitable comparison of the anode's effective capacity contribution within full-cell configurations.)

Modifications	Cathode	Rate	Cycling Life	Total Specific Capacity of Anode	Capacity Decay Rate	Ref.
Li ₂ ZrF ₆ -based electrolytes	LFP	1C	220 cycles	230 mA h g ⁻¹	0.091%	Nature 637 , 339–346 (2025)
PFB electrolyte with a large oscillatory degree	NCM811	0.2C	150 cycles	554 mA h g ⁻¹	0.053%	Nat. Energy , 9 , 1285–1296 (2024)
	NCM613	0.5C	400 cycles	681 mA h g ⁻¹	0.049%	
Bilayer structure of SEI tailored through trioxane-modulated electrolytes	NCM811	0.2C	130 cycles	804 mA h g ⁻¹	0.064%	Nat. Energy , 8 , 725–735 (2023)
A compatible electrolyte and uniform external pressure	NCM622	0.3C	200 cycles	635 mA h g ⁻¹	0.070%	Nat. Energy , 4 , 551–559 (2019)
Solid-Solution Li-Ag alloy	NCM811	0.5C	250 cycles	337 mA h g ⁻¹	0.052%	J. Am. Chem. Soc. 145 , 24775–24784 (2023)
Anion-receptor-mediated carbonate electrolyte	NCM811	0.2C	50 cycles	570 mA h g ⁻¹	0.182%	Angew. Chem. Int. Ed. 60 , 19232–19240 (2021)
Li/C composite anode	NCM523	0.1C	150 cycles	391 mA h g ⁻¹	0.135%	J. Am. Chem. Soc. 144 , 212–218 (2022)
Dual-Layered Artificial Interphase	NCM613	0.5C	60 cycles	391 mA h g ⁻¹	0.152%	Adv. Mater. 35 , 2300350 (2023)
3D Li/Li ₂₂ Sn ₅	NCM622	0.5C	45 cycles	295 mA h g ⁻¹	0.207%	Adv. Energy Mater. 13 , 2302755 (2023)
Tailoring solvation structures via precise diluent engineering	NCM811	1C	107 cycles	583 mA h g ⁻¹	0.075%	Adv. Mater. e09109 (2025)
Liquid–liquid interfacial tension	NCM811	0.1C	189 cycles	641 mA h g ⁻¹	0.101%	Nature 643 , 1255–1262 (2025)
This work	NCM811	0.5C	300 cycles	947 mA h g⁻¹	0.028%	

Comment (3): It would be better to provide a deeper analysis of long-term electrochemical stability, specifically detailing the mechanism of Li deposition and interface evolution with quantitative in-depth characterization.

Responses to Comment 3: We sincerely appreciate the reviewer's insightful suggestion. To elucidate the relationship between cycling durability and structural/interface evolution, we have conducted a series of *in situ*, post-mortem characterizations on the electrochemical process of LZNC anode and bare Li anode upon the long-term cycling complemented by systematic, quantitative analysis.

1) *In Situ* Visualization of Li Deposition/Stripping Behavior of LZNC

In situ optical microscopy observations of symmetric cells (Fig. R14) provide direct visual evidence of the lithium deposition and stripping behaviors between the LZNC, bare Li, and Li-Zn electrodes. Electrochemical testing was performed at a current density of 1 mA cm^{-2} , with an areal capacity of 2 mA h cm^{-2} , the LZNC electrode demonstrates exceptional structural integrity without fracture or pulverization. This preserved 3D conductive framework subsequently guides uniform lithium replating, resulting in dense, homogeneous Li deposition devoid of dendritic morphologies. Conversely, control electrodes comprising bare Li and Li-Zn alloy suffer from severe structural degradation during stripping, characterized by the formation of numerous voids and a progressively looser, more porous architecture. This structure degradation induces irregular, localized lithium nucleation upon subsequent plating, with formation of dendrite or mossy Li.

Fig. 14. The *in situ* optical observations of Li stripping/plating processes on Li, Li-Zn and LZNC electrodes at a current density of 1 mA cm^{-2} .

2) Post-Cycling Structural Characterization of LZNC

Ex situ scanning electron microscopy (SEM) analysis of electrodes cycled to various stages (50, 100, and 300 cycles; Fig. R15) reveals a fundamental divergence in morphological evolution between the LZNC and bare lithium electrodes. The LZNC electrode maintains a remarkably compact and uniform surface morphology throughout extended cycling, with no evidence of cracking, pulverization, or dendritic formation. This structural preservation indicates exceptionally homogeneous lithium deposition and stripping, facilitated by the stable conductive scaffold. In contrast, the bare lithium electrode undergoes severe and progressive structural degradation from 50 to 300 cycles, developing a rough surface dominated by porous, mossy lithium deposits. In addition, a substantial crack between a top layer of freshly deposited lithium and the underlying unreacted lithium metal was observed for bare Li after 300 cycles. This microstructural failure blocks ionic and electronic pathways, directly accounting for the increasing polarization and eventual cell failure observed in electrochemical data. Cross-sectional SEM images further confirm the preservation of the integrated bulk framework of LZNC without structural collapse, whereas bare lithium exhibits significant structural degradation, characterized by massive cracks and the accumulation of inactive (dead) lithium.

The exceptional structural stability of the LZNC electrode stems from its unique resistance to pulverization, which enables the preservation of the original 3D ionic and electronic conductive framework even after complete stripping of the metallic lithium, as verified by X-ray microscopy (XRM) results (Fig. R16).

Fig. R15. SEM images of LZNC and Li foil surfaces at different cycle stages.

Fig. R16. a-d) 3D microstructural analysis of LZNC after Li stripping and e-h) x-y slices detected by XRM.

3) Quantitative Analysis of Interfacial Kinetics and Degradation of Anodes

EIS (Figs. R17 and R18) provides quantitative insight into the interfacial evolution and kinetics of the LZNC electrode during extended cycling (Figs. R14 and R15). Comparative measurements were conducted using both LZNC and pure lithium anodes cycled under identical conditions (1 mA cm^{-2} , 1 mA h cm^{-2}) over 20, 40, 60, 80 and 100 cycles. The LZNC composite consistently exhibits significantly lower interfacial (R_s) and charge-transfer resistance (R_{ct}) compared to bare lithium throughout cycling, indicating the stabilization of the electrode-electrolyte interface and highly efficient ionic transport. Notably, the interfacial resistance (R_s and R_{ct}) of LZNC decreases during the initial activation and subsequently stabilizes, indicating the formation of a robust and rapid ionic-transfer conductive interface. In contrast, the bare Li electrode showed a sharp increase in both R_s and R_{ct} after approximately 80 cycles, following initial activation period. This rapid impedance increase is directly attributed to severe structural degradation and repeated solid electrolyte interphase (SEI) fracture.

Fig. R17. Nyquist plots of LZNC and Li foil at different cycle stages.

Fig. R18. Bar charts comparing R_{CT} and R_s obtained from EIS spectrum fitting of LZNC and Li foil at different cycle stages.

4) Chemical Composition and Evolution of the Anode Interface

To elucidate the chemical composition and its dynamic evolution at the interface, we further performed ex situ X-ray photoelectron spectroscopy (XPS) on those cycled LZNC anodes (Figs. R19-R20). The spectra confirm the persistent presence of both Li_3N and Li-Zn phases at the electrode interface throughout extended cycling. The highly ionic conductive Li_3N facilitates rapid Li^+ transport, while the lithiophilic Li-Zn alloy serves as a homogenous nucleation site, which facilitate to guide uniform Li deposition. Quantitative deconvolution of the XPS spectra reveals a pronounced evolution in SEI composition with cycling. We observe a progressive enrichment of inorganic species, predominantly Li_3N , accompanied by a decrease of organic species. Notably, the observation of a characteristic C-N peak in the XPS spectrum after 200 cycles confirms the continuous enrichment of Li_3N on the CNTs surface during the cycling process. This compositional evolution indicates the formation of a stable, inorganic-dominated SEI, which enhances both interfacial ion-transport kinetics and mechanical stability.

Fig. R19. XPS profile of N 1s and Zn 2p for LZNC at different cycle stages.

Fig. R20. Comparison of atomic percentages on the surface of LZNC anodes at different cycle stages from XPS measurement results.

5) Mechanical Properties of the Anode Interface

Atomic force microscopy (AFM) measurements (Figs. R21 and R22) provide quantitative insight into the mechanical evolution of the electrode interface. The surface modulus of the LZNC electrode increases with progressive cycling, which is consistently higher than that of bare lithium throughout extended operation. This mechanical enhancement is directly correlated with the formation of an inorganic-rich interphase, as confirmed by XPS analysis. Such mechanical reinforcement is critical for inhibiting lithium dendrite penetration and preserving structure stability. Consequently, even after 300 cycles, the LZNC electrode maintains an exceptionally smooth surface morphology, indicative of uniform, dense lithium deposition (refer to Figs. R15 and R21). In contrast, cycled bare lithium electrodes exhibit severe surface roughening, characterized by irregular dendrites and a substantial accumulation of electrochemically isolated (dead) lithium. This pronounced divergence in interfacial integrity and morphological stability highlights the essential role of a mechanically robust, inorganic-stabilized interphase in achieving long-term cycling performance.

In conclusion, this comprehensive suite of experimental techniques, including *in situ* morphological monitoring, *ex situ* SEM, EIS, XPS analysis, and AFM characterization, provides a comprehensive mechanistic understanding of the exceptional long-term performance of LZNC anode. The superior cycling stability originates from the synergistic combination of structural integrity, maintained by the highly-toughness scaffold, and electrochemical interface stability, enabled by the

inorganic-enriched and mechanical robustness. These properties work in concert to ensure dense, dendrite-free lithium plating/stripping behavior, ultimately enabling remarkable cycle life under practical operating conditions.

In the revised manuscript, we have updated figures (*Text of Figs. 4f, 4g, 5i, S20, S24, S30, S36, S37 and S38*) and included more discussion (*Page 7, 10, 12 and 13*).

Fig. R21. AFM modulus of LZNC and Li foil at different cycle stages.

Fig. R22. Comparison of AFM modulus values for LZNC and Li foil at different cycle stages.

Comment (4): The electrochemical and structural characterizations, while extensive, still lack critical in-situ insights into the structural dynamics during actual battery operation conditions.

Responses to Comment 4: We sincerely thank the reviewer for this insightful comment. We fully agree that *in situ* and operando characterizations are indispensable for elucidating the dynamic structural and interfacial evolution of battery materials under operating conditions. In response to this point, our revised manuscript now incorporates a series of complementary operando and quasi-*in situ* techniques designed to provide a

holistic, real-time view of the (de)lithiation mechanisms within the LZNC composite anode.

1) Operando Optical Microscopy

This technique provides direct, real-time visualization of Li deposition and stripping dynamics. As shown in Fig. R12, during the Li stripping at 1 mA cm^{-2} with a capacity of 2 mA h cm^{-2} , the LZNC electrode maintains a continuous, intact three-dimensional skeleton. In contrast, both the LiZn alloy and bare Li electrodes exhibit severe structural degradation, characterized by observable cracking looser and more loosen architecture. Upon subsequent Li plating, lithium is homogeneously deposited within the robust LZNC scaffold, with no evidence of dendritic growth. Conversely, lithium deposition on the Li-Zn and bare Li electrodes is highly irregular and occurs predominantly on the surface, accompanied by extensive dendrite formation.

2) Operando X-ray Diffraction (XRD)

To dynamically probe the crystallographic phase evolution, we conducted operando XRD on the LZNC electrode during galvanostatic cycling at 0.1 mA cm^{-2} (Fig. R23). It reveals a stepwise delithiation mechanism. The process initiates with the removal of free metallic Li, followed by the partial dealloying of the Li-Zn alloy, with CNTs framework remains structurally intact throughout the process. Upon Li plating, the Li-Zn alloying reaction occurs initially, followed by the uniform deposition of metallic lithium within the matrix. This result highlights the chemical reversibility and structural robustness of the composite.

Fig. R23. *In-situ* XRD analysis and schematic illustration of the Li stripping and plating processes of LZNC anode

3) Operando EIS

Operando EIS was employed to track the real-time evolution of interfacial resistance during cycling at 0.5 mA cm^{-2} with a fixed capacity of 5 mA h cm^{-2} (Fig. R24). For LZNC batteries, the interfacial resistance (R_s) gradually decreases throughout plating/stripping cycles, indicative of a persistent stabilized SEI and highly efficient charge transfer within the 3D framework. In contrast, the LiZn electrode exhibits significant and erratic fluctuations in impedance, which could be attributed to repetitive SEI fracture resulting from the mechanical collapse of the framework and uncontrolled dendritic lithium deposition, as conclusively verified by post-mortem SEM analysis. Noteworthy, the impedance for LiZn remained consistently higher than that of LZNC throughout the entire process.

Fig. R24. *In-situ* Distribution of Relaxation Times (DRT) analysis of the Li stripping and plating processes of Li-Zn and LZNC anode.

4) Quasi-*In Situ* XPS

The exceptional long-term cyclability of LZNC is fundamentally facilitated by the chemical and electrochemical stability of its interfacial components. To quantitatively probe this, we performed quasi-*in situ* XPS analysis on LZNC electrode at various Li stripping states (cut-off voltages of 0.1, 0.2, 0.3, 0.4, 0.5, 0.6, 0.7, 0.8, 0.9 and 1.0 V; Fig. R25). The high-resolution spectra reveals that the characteristic signal of Li_3N and metallic Zn (Li-Zn) remain essentially unchanged in both binding energy and peak intensity. This demonstrates the exceptional chemical stability of the interface of LZNC throughout the electrochemical cycling process.

Fig. R25. Quasi-*in situ* XPS analysis of changes in N 1s and Zn 2p during Li stripping of LZNC anodes.

5) *In-situ* analysis of thickness expansion behavior in pouch cells

To further probe the structural dynamics of the anodes under practical operating conditions, we conducted *in-situ* thickness variation measurements using *in-situ* cell swelling analyzer employing bare Li and LZNC anodes paired with NCM811 cathodes. Detailed experimental conditions are provided in the Supporting Information. This measurement enables direct and quantitative assessment of electrode volume evolution during repeated lithium deposition and stripping processes. As shown in Figure R26, the NCM811||LZNC pouch cell exhibits a significantly lower thickness fluctuation throughout cycling compared to the NCM811||Li cell. The reduced thickness variation indicates a highly uniform, reversible lithium deposition/stripping process within the LZNC framework. This behavior demonstrates that the 3D conductive and mechanically robust LZNC matrix can effectively accommodate the strain induced by lithium depositions/stripping and mitigate localized volume expansion. These observations collectively highlight that the LZNC anode enables more homogeneous electrochemical reactions and enhanced structural integrity in practical full-cell configurations, supporting its practical applicability in high-energy-density lithium metal batteries.

Fig. R26. Thickness variation of NCM811||LZNC and NCM811||Li pouch cell during charging and discharging process.

In conclusion, these multi-modal operando analyses provide a comprehensive investigation of the LZNC anode upon battery operation. It reveals that the LZNC composite facilitates uniform lithium deposition, maintains mechanical integrity of its 3D host, and promotes the formation of a stable, low-impedance interface, thereby accounting for its superior electrochemical performance.

In the revised manuscript, we have updated figures (*Text of Figs. 3a, 3b, 3c, S20, S21 and S46*) and included more discussion (*Page 7 and 14*).

Comment (5): Post-mortem characterization of the cycled electrodes is necessary to support claims regarding durability and interfacial stability.

Responses to Comment 5: We sincerely thank the reviewer for this valuable suggestion. We fully agree that systematic post-mortem characterization is essential to substantiate the claimed durability and interfacial stability of the LZNC composite anode. In response, we have conducted a comprehensive multi-technique analysis of cycled electrodes to provide conclusive evidence of structural and interfacial robustness.

1) Structural Integrity and Morphological Stability

To investigate the evolution of structural and morphological of LZNC upon cycling, we performed complementary *Optical microscopy, XRM, and SEM* analysis on the cycled electrodes (Fig. R27).

Optical microscopy performed at various depths of delithiation (25% and 75%; Fig. R27a-d) demonstrates that the LZNC electrode maintains its structural integrity even under deep stripping conditions, whereas a control Li-Zn alloy suffers rapid

fracture and pulverization. *XRM* corroborates these findings, revealing the preservation of a continuous 3D Zn-CNTs network even after complete delithiation (100% delithiation). This structural resilience originates from the inherent mechanical robustness of the composite architecture, ensuring electrode integrity under practical cycling conditions. *Post-cycling SEM* characterization of the LZNC anode after 300 cycles (Fig. R27) reveals a compact and uniform surface morphology with no evidence of cracking, pulverization, or dendritic growth. Cross-sectional images further confirm the preservation of the integrated bulk framework without structural collapse, whereas bare lithium exhibits significant structural degradation, characterized by massive cracks and the accumulation of inactive (dead) lithium. We also conducted *ex situ* SEM characterization of LZNC and bare Li cycled at different cycling stages, which confirms LZNC maintains a remarkably compact and uniform morphology throughout extended cycling, whereas bare lithium undergoes severe and progressive structural degradation (refer to *Response to Comment 3*, Fig. R15).

Fig. R27. a-d) Digital photograph of Li-Zn and LZNC stripped 25% and 75% free Li. e-h) 3D microstructural analysis of LZNC after Li stripping and corresponding x-y

slices detected by XRM. i-l) SEM images of the surface and cross-section of LZNC after 300 cycles.

2) Interfacial Chemistry and Mechanical Properties

To investigate the evolution of interfacial chemistry and mechanical properties, we performed *XPS in-depth profiling*, *AFM mapping*, and *Time-of-flight secondary ion mass spectrometry* analysis on the cycled electrodes (Fig. R28).

XPS depth profiling of cycled LZNC electrodes revealed a distinct contrast with bare lithium. A strong signal from carbon-containing organic compounds, derived from electrolyte solvent decomposition, was detected on the bare Li surface (Figs. R28a-d). In contrast, the LZNC surface exhibited a relatively weak C 1s signal (Figs. R28a-d). Furthermore, with increasing etching depth, the carbon signal decreased significantly while the intensities of nitrogen (N) and zinc (Zn) remained constant (Figs. 5f-h), indicating that the interface is dominated by inorganic species both at the surface and within the bulk.

Ex situ XPS analysis at different cycling stages confirmed the persistent presence of both Li_3N and Li-Zn phases at the electrode interface. Quantitative spectral deconvolution revealed a dynamic evolution in SEI composition, characterized by a progressive enrichment of inorganic species (predominantly Li_3N , refer to *Response to Comment 3*, Figs. R19-20). This continuous formation of Li_3N was further corroborated by the emergence of a characteristic C-N peak after 200 cycles.

The mechanical implications of this inorganic-rich interphase were validated by **AFM mapping**. The LZNC electrode exhibited a substantially higher surface modulus (13.6 GPa) compared to bare lithium (387 MPa). This mechanical superiority was maintained throughout extended cycling (refer to *Response to Comment 3*, Figs. R21-22), with the LZNC's surface modulus increasing progressively, a trend directly correlated with the inorganic species enrichment observed via XPS.

TOF-SIMS data provided 3D compositional validation (Figs. R28i-k), demonstrating a homogeneous and stable distribution of key inorganic species (Li, N, Zn). This confirms the formation of a stable, dendrite-suppressing interface with

favorable ion-transport properties.

Fig. R28. a) Comparison of atomic percentage on the surface of bare Li and LZNC anodes from in-depth XPS measurements. XPS depth profile of b) C 1s, c) Zn 2p, and d) N 1s of LZNC anode after cycled for 50 cycles. AFM modulus and morphology comparison of SEI for e, g) bare Li and f, h) LZNC anodes after cycled for 50 cycles. i-k) TOF-SIMS analysis of the LZNC anode after 300 cycles.

3) Charge-Transfer Kinetics and Interfacial Stability

To investigate the charge-transfer kinetics and the interfacial stability upon cycling, we performed *EIS analysis and Tafel measurements* on the cycled electrodes.

EIS analysis confirms that the LZNC electrode exhibits lower and stable interfacial (R_s) and charge-transfer resistance (R_{ct}) throughout extended cycling, indicating minimal degradation and rapid interfacial kinetics (refer to *Response to Comment 3*, Figs. R17 and R18). *Tafel measurements* on batteries after multiple cycles reveal a notably higher exchange current density for LZNC ($0.0887 \text{ mA cm}^{-2}$) compared

to Li-Zn ($0.0131 \text{ mA cm}^{-2}$) and bare lithium (0.011 mA cm^{-2}), attributable to the biphasic Li_3N -CNTs network that facilitates concurrent ion and electron transport (refer to *Response to Comment 1*, Fig. R9).

These post-mortem findings are fully consistent with the *in situ* and operando characterizations detailed in *Response to Comment 4*. These comprehensive post-mortem analysis presented herein robustly supports the conclusion that the LZNC composite exhibits superior structural and interfacial durability under practical, long-term cycling conditions.

In the revised manuscript, we have updated figures (*Text of Figs. 5, S22, S24 and S36*) and included more discussion (*Page 7 and 12*).

For Reviewer #3

General Comments: The three-dimensional lithium metal composite anode (LZNC) proposed in this study aims to resolve the contradictions between mechanical toughness, processability, structural integrity, and electrochemical performance of ultrathin lithium metal anodes by synergistically integrating Li-Zn alloy with Li₃N-enriched carbon nanotubes (Li₃N-CNTs). However, there are some unclear points in this manuscript, with a few flaws in the experimental data. Furthermore, the innovative aspects of this research have not been fully explored. Based on these considerations, major revisions are required before its publication in *Nature Communications*.

Response to General Comments: We sincerely thank the reviewer for their comprehensive assessment and valuable suggestions. In response to the general comments, we have thoroughly revised the manuscript to strengthen the clarity, depth, and novelty of our work. The core innovation of this study lies not merely in the combination of materials, but in the rational multi-scale design of a nanocomposite foil (LZNC) that introduces a toughness-dominated paradigm for ultra-thin lithium metal anodes. By synergistically integrating a lithiophilic Li-Zn alloy phase with a Li₃N-enriched carbon nanotube (Li₃N-CNTs) network, the LZNC anode simultaneously addresses the critical trade-offs among mechanical robustness, processability, structural integrity, and electrochemical performance that have long hindered conventional anode designs. The superiority of this approach is unambiguously demonstrated through three key advances as follows.

1) A Shift in Mechanical Design of Foil Anodes: from Strength to Toughness

Conventional lithium composite anodes predominantly emphasize enhancing mechanical strength (defined as the maximum stress before fracture). In this work, we introduce and prioritize high toughness, the energy absorbed per unit volume before fracture (J/m³), as a critical design metric for freestanding lithium anode foils. This is achieved through a hierarchical architecture in which the dispersed Li-Zn intermetallic phase serves as a strengthening component, while the interwoven, elastic Li₃N-CNT network provides exceptional ductility and crack-bridging capability. This synergistic

effect endows the LZNC foil with a tensile strength of 23.9 MPa and a remarkable rupture toughness of $1.3 \times 10^6 \text{ J/m}^3$, representing an approximately 12-fold improvement over bare lithium (Fig. R29). As a result, the LZNC anode can be processed into ultra-thin foils ($\sim 8 \text{ }\mu\text{m}$) to increase cell-level specific energy. Moreover, the high toughness enables the foil to intrinsically resist crack propagation and structural pulverization even under deep stripping conditions, ensuring long-term cycling stability under demanding operational scenarios.

Figure R29. a) Tensile stress-displacement profiles of Li, Li-Zn, LZN, LZC and the LZNC foil, with the integrated area beneath the stress-strain curve (shaded region) quantifying its fracture toughness. b) Columnar Comparison Chart of Tensile Strength and Toughness for Li, Li-Zn, LZN, LZC, and LZNC.

2) Reconciliation of Ultra-Thin Processability with High Gravimetric and Volumetric Capacity

The pursuit of high-energy-density batteries necessitates the development of ultra-thin, lithium-less anodes to minimize the use of metallic lithium. However, the fabrication of ultrathin Li anodes presents a fundamental challenge. Pure lithium metal is difficult to roll thinly due to its high viscosity and low mechanical strength, while many 3D host-based composites (e.g., Li-C) exhibit brittleness and low volumetric capacity owing to excessive porosity (Table R2, Fig. R30). Although alloy foils (e.g., Li-Mg, Li-Al) can be processed thinly, they typically lack a continuous supporting matrix and suffer from structural pulverization upon dealloying, often requiring heavy current collectors that severely compromise gravimetric capacity. Our LZNC anode overcomes these limitations. Its inherent ductility and toughness enable the fabrication

of current-collector-free foils below 8 μm . Crucially, the dense composite structure delivers a high gravimetric capacity of 1800 mA h g^{-1} and an exceptional volumetric capacity of 1915 mA h cm^{-3} , substantially exceeding most reported Li-C and Li-alloy composites (Table R2, Fig. R30).

Fig. R30. Comparison of volumetric/gravimetric specific capacity of the previously reported 3D Li-C and 3D Li-alloy anodes. Note that the circumferential area inversely correlates with the thickness of the composite anodes.^[1-14] Detailed parameters are provided in Supplementary Table 2.

Table R2. Comparison of volume and gravimetric specific capacity for reported composite anode.

	Composite material	Gravimetric specific capacity (mA h g^{-1})	Volume specific capacity (mA h cm^{-3})	Thickness (μm)	Ref.
Li-C	Li/3D hollow carbon fiber	1107	363	165	Joule 1 , 563–575 (2017)
	Li/hierarchical silver-nanowire-Graphene	1403	342	350	Adv. Mater. 30 , 1804165 (2018)
	Li/CNTs-MC	1640	793	126	Adv. Mater. 31 , 1805654 (2019)
	Li/coralloid carbon fiber	1570	666	150	Joule 2 , 764–777 (2018)
	Li@GDD-CH	1598	652	46	Angew. Chem. Int. Ed. 63 , e202403399 (2024)
	ZOS-CF@Li	1115	584	137	Adv. Funct. Mater. 35 , 2420382 (2025)
Li-Alloy	Li/Li ₂₂ Sn ₅ /LiF	606	1599	104	Adv. Mater. 35 , 2207310 (2023)
	Al-HCGB-Li	968	1776	50	Sci. Adv. 8 , eabq3445 (2022)
	Li/Li ₂₂ Sn ₅	656	841	400	Nat. Commun. 11 , 829 (2020)
	LiMg/CuCM	1401	633	104	Adv. Mater. 34 , 2205677 (2022)
	Li-B@SSM	996	1445	98.35	Adv. Mater. 35 , 2211203 (2023)
	Fe-N@SSM-Li	628	571	105	Adv. Funct. Mater. 33 , 2308022 (2023)
	3D Li/Li ₂₂ Sn ₅	1178	1798	100	Adv. Energy Mater. 13 , 2302755 (2023)
	TFA	1078	1552	100	Adv. Mater. 37 , 2506298 (2025)
	This work	1799	1915	8	

3) Simultaneous Realization of Long Cycle Life and High Energy Density in

Practical Pouch Cells

The ultimate validation of an anode lies in its performance in full-cells under practical conditions, where achieving both long cycle life and high energy density remains particularly challenging. The mechanical robustness and continuous ion/electron-conducting network of the LZNC anode enable superior comprehensive full-cell performance. To ensure a fair comparison with state-of-the-art systems, we introduce a *standardized anode specific capacity* metric (defined as $NSC = \frac{Capacity_{anode}}{(mass_{anode} + mass_{current\ collector}) \times N/P}$), which normalizes effective capacity by accounting for the mass of any integrated current collector and the N/P ratio.

Under this rigorous assessment, the NCM811||LZNC pouch cell outperforms previously reported systems, achieving a record-high NSC of 947 mA h g^{-1} , as well as prolonged cycle life and an outstanding per-cycle capacity retention of 99.94%. In contrast to prior studies that frequently emphasize either high (anode) capacity or long cycle life in isolation, the LZNC anode delivers both simultaneously under realistic pouch-cell conditions, benchmarking favorably against the most advanced Li-metal systems reported (Table R3, Fig. R31). Furthermore, an 8.5 Ah pouch cell demonstrates a practical energy density of 553 W h kg^{-1} , confirming its high-specific-energy potential.

Fig. R33. Comparison of normalized total anode specific capacity versus working cycle life in the previously reported Li-metal pouch cells. Note that the circumferential area inversely correlates with the capacity decay rate of the batteries. ^[15-22] Detailed

parameters are provided in Supplementary Table 5.

Table R3. Comparison of cycling performance for previously reported pouch cells. (Total specific capacity of anode refers to the ratio of the capacity of the cell exerted in cycling to the mass of the complete anode including the collector.)

Modifications	Cathode	Rate	Cycling Life	Total Specific Capacity of Anode	Capacity Decay Rate	Ref.
Li ₂ ZrF ₆ -based electrolytes	LFP	1C	220 cycles	230 mA h g ⁻¹	0.091%	Nature 637 , 339–346 (2025)
PFB electrolyte with a large oscillatory degree	NCM811	0.2C	150 cycles	554 mA h g ⁻¹	0.053%	Nat. Energy , 9 , 1285–1296 (2024)
	NCM613	0.5C	400 cycles	681 mA h g ⁻¹	0.049%	
Bilayer structure of SEI tailored through trioxane-modulated electrolytes	NCM811	0.2C	130 cycles	804 mA h g ⁻¹	0.064%	Nat. Energy , 8 , 725–735 (2023)
A compatible electrolyte and uniform external pressure	NCM622	0.3C	200 cycles	635 mA h g ⁻¹	0.070%	Nat. Energy , 4 , 551–559 (2019)
Solid-Solution Li-Ag alloy	NCM811	0.5C	250 cycles	337 mA h g ⁻¹	0.052%	J. Am. Chem. Soc. 145 , 24775–24784 (2023)
Anion-receptor-mediated carbonate electrolyte	NCM811	0.2C	50 cycles	570 mA h g ⁻¹	0.182%	Angew. Chem. Int. Ed. 60 , 19232–19240 (2021)
Li/C composite anode	NCM523	0.1C	150 cycles	391 mA h g ⁻¹	0.135%	J. Am. Chem. Soc. 144 , 212–218 (2022)
Dual-Layered Artificial Interphase	NCM613	0.5C	60 cycles	391 mA h g ⁻¹	0.152%	Adv. Mater. 35 , 2300350 (2023)
3D Li/Li ₂₂ Sn ₅	NCM622	0.5C	45 cycles	295 mA h g ⁻¹	0.207%	Adv. Energy Mater. 13 , 2302755 (2023)
Tailoring solvation structures via precise diluent engineering	NCM811	1C	107 cycles	583 mA h g ⁻¹	0.075%	Adv. Mater. e09109 (2025)
Liquid–liquid interfacial tension	NCM811	0.1C	189 cycles	641 mA h g ⁻¹	0.101%	Nature 643 , 1255–1262 (2025)
This work	NCM811	0.5C	300 cycles	947 mA h g⁻¹	0.028%	

Figure R32. The radar chart displays the key performance characteristics of Li, Li-C, Li-alloy, and LZNC.

In summary, this work moves beyond conventional approaches focused on isolated properties by pioneering a multi-dimensional design framework that

synergistically optimizes mechanical, dynamic, and processing characteristics. The LZNC foil represents an integrated anode architecture that simultaneously achieves rapid ion/electron transport, scalable processability, mechanical toughness, and high volumetric/gravimetric capacity. We contend that the design principles established herein provide a new pathway for developing high-energy, long-cycle-life lithium metal batteries. In the revised manuscript, we have re-structured the Abstract, Introduction, and Conclusion to clearly highlight these innovations.

Comment (1): As can be seen from Figure 1a, the authors added CNTs during the smelting process. Then, at high temperatures, why did CNTs not react with Li to form Li-C_x ? If Li-C_x was formed, would it change the fibrous structure of CNTs? If the fibrous structure of CNTs were not changed, more intuitive morphological structure characterization should be provided.

Responses to Comment 1: We appreciate the reviewer's insightful questions concerning the chemical and structural stability of CNTs upon exposure to molten lithium. Our comprehensive characterizations confirm that lithiation of CNTs does occur, but through a well-defined, modest intercalation mechanism that preserves the essential fibrous morphology of the CNTs.

The formation of lithium-intercalated carbon (LiC_x) is unequivocally demonstrated through XRD and XPS analysis. The (002) diffraction peak, corresponding to the graphitic stacking in pristine CNTs, shifts from 25.92° to 23.85° after composite fabrication. ^[25,26] This signifies an expansion of the interlayer spacing from approximately 3.57 \AA to 3.73 \AA , a hallmark of lithium intercalation into the graphitic layers. ^[27,28] Consistent with this, prior research by Shao et al. ^[29] illustrated the lithiation of graphite coatings by molten lithium in solid electrolyte, identifying structural defects, voids, and CNTs ends as preferential nucleation points for the reaction. This finding is consistent with observations over other graphite-based materials. ^[30] The reason for the relatively weak scattering intensity is attributed to their low crystallinity of the carbonaceous materials compared to the highly crystalline metal/alloy phases present in the LZNC. Furthermore, the carbon spectrum from XPS

result of the LZNC composite reveals a new peak at 282.9 eV, which we assign to Li-C bonds (Fig. R33), providing direct chemical evidence for the formation of lithiated-CNTs.

Critically, this intercalation process does not compromise the structural integrity of the CNT network. To directly assess morphology, we performed ex-situ SEM on a partially delithiated LZNC sample (electrochemically charged to 0.1 V vs. Li⁺/Li to remove free lithium metal while retaining the lithiated CNTs framework). The result (Fig. R34) clearly shows that the CNTs maintain their original tubular morphology and form a continuous, interwoven network without collapse or fracture. Furthermore, CNTs are widely utilized as host materials for lithium metal anodes due to their 3D interconnected network, and prior reports have consistently shown that lithiated CNTs retain their original tubular morphology, which aligns with our experimental observations.^[31] Therefore, while the CNTs are indeed lithiated to form LiC_x, this process is fundamentally an intercalation reaction that expands the lattice but crucially maintains the continuous, fibrous scaffold.

In the revised manuscript, we have included more discussion in the text and Supplementary materials to clarify the point (*Text of Figs. S13 and S14*) and updated the manuscript accordingly (*Page 6*).

Fig. R33. a) XRD pattern of LZNC. b) XRD patterns of pristine CNTs and LZNC, showing an enlarged view of the region corresponding to the dashed rectangle in (a). c) XPS spectra of C 1s of the LZNC composite.

Fig. R34. SEM image of LZNC electrode after charging at 0.1 V vs. Li^+/Li , a process that removes free lithium metal while preserving the lithium-intercalated carbon nanotube framework.

Comment (2): As mentioned by the authors in Figure 3a, "the voltage gradually rises while the peak intensity of LiZn diminishes, accompanied by the emergence of the peak of Zn ". It is crucial to clarify whether this dealloying process is reversible, which is highly relevant to the structural integrity of the 3D framework during lithium deposition/stripping with large areal capacities.

Responses to Comment 2: We thank the reviewer for raising this critical point regarding the reversibility of the dealloying process, which is indeed fundamental to the structural and functional integrity of the 3D framework during cycling. Our *in-situ* XRD data (Fig. 3a) provides direct evidence for the complete reversibility of this process. As shown in Fig. 3a, a Zn diffraction peak emerges with the diminishment of the LiZn peak intensity during delithiation, while during subsequent lithiation, the Zn signal vanishes as the LiZn reflection recovers, which confirms the complete reversibility of the dealloying process. Notably, the LiZn peaks exhibit no cumulative broadening upon cycling, indicating the structure stability of the alloy framework without structural degradation. To further corroborate the morphological reversibility and structural integrity of the dealloyed framework, we performed *ex-situ* SEM and EDS mapping on a sample charged to 1.0 V (vs. Li^+/Li), a potential at which delithiation is complete. These analyses reveal a continuous, interwoven network where Zn domains exhibit an island-like morphology percolated by CNTs matrix, with no evidence of structural fragmentation or cracking (Fig. R35). This morphological integrity was further confirmed by XRM (Fig. R36), which visually affirms that the Zn

particles remain interconnected and well-anchored by the CNTs network after complete lithium stripping.

The stability of this 3D framework enables excellent cycling performance of LZNC-based full-cells under practical conditions. To demonstrate this, we paired the LZNC anode with a high-loading $\text{LiNi}_{0.8}\text{Co}_{0.1}\text{Mn}_{0.1}\text{O}_2$ (NCM811) cathode (areal capacity: 4 mA h cm^{-2}) at a low N/P ratio of 1.9. The full cell retained 92% of its capacity after 300 cycles at 0.5C, highlighting the exceptional structural reversibility of the LZNC composite even under demanding cycling conditions.

In the revised manuscript, we have included more discussion in the text and Supplementary materials to clarify the point (*Text of Figs. 3d and S23*) and updated the manuscript accordingly (*Page 7*).

Fig. R35. SEM image and corresponding EDS mapping of LZNC after complete delithiation.

Fig. R36. X-ray CT reconstruction of the fully delithiated LZNC anode.

Comment (3): According to Figure 6f and Figure S33, it can be concluded that the

assembled pouch cell has a capacity of 7.6 Ah and a mass of 0.058 kg. The median voltage of NCM811 is generally 3.8 V, and based on this calculation, the overall energy density is only 497.9 Wh/kg. However, the data calculated in the manuscript is 553 Wh/kg. A reasonable explanation should be provided.

Responses to Comment 3: We thank the reviewer for their meticulous examination of our energy density calculation. The discrepancy arises from the use of standard $\text{LiNi}_{0.8}\text{Co}_{0.1}\text{Mn}_{0.1}\text{O}_2$ (NCM811) parameters in the reviewer's estimation, whereas our pouch cell utilizes a $\text{LiNi}_{0.98}\text{Co}_{0.01}\text{Mn}_{0.01}\text{O}_2$ cathode with much higher nickel content. This cathode chemistry operates within a wider voltage window (2.5- 4.5 V), leveraging its high-voltage phase characteristics to deliver a higher average discharge voltage and greater specific capacity than conventional NCM811.

Importantly, the capacity of 7.6 Ah mentioned in the manuscript was a preliminary estimated value based on the theoretical specific capacity of $\text{LiNi}_{0.98}\text{Co}_{0.01}\text{Mn}_{0.01}\text{O}_2$ cathode materials and the designed electrode loading before cell testing. This estimated value deviates from the actual measured value. The actual measured discharge capacity of the assembled pouch cell, as shown in the capacity-voltage profile (Fig. R37), reaches 8.5 Ah. More critically, the energy-voltage curve (obtained from the testing software) yields a total energy output of 32.36 Wh. Given the total cell mass of 58.35 g, the specific energy of this pouch cell is calculated to be of 553 Wh kg^{-1} . To avoid misunderstanding, we have corrected the estimated pouch cell capacity to the experimentally measured value (8.5 Ah) in the revised manuscript.

To ensure transparency and facilitate comparison, we have provided a detailed summary of the pouch cell configuration, including areal capacity, electrode loadings, active material utilization, and total mass, in Table R4. This allows for direct verification of both the practically measured capacity and the resulting specific energy. In the revised manuscript, we have included more discussion in the Supplementary materials to clarify the point (*Text of Fig. S47 and Table S7*).

Fig. R37. a) Capacity-voltage, b) energy-voltage profiles and c) optical photograph of the 8.5 Ah LiNi_{0.98}Co_{0.01}Mn_{0.01}O₂||LZNC pouch cell.

Table R4. Parameters of the 8.5A h LiNi_{0.98}Co_{0.01}Mn_{0.01}O₂||LZNC pouch cell.

Composition	Parameter	Value
Ni98 Cathode	Areal capacity (mA h cm ⁻²)	11.8
	Mass loading (mg cm ⁻²)	27.5
	Stacked number of layers	15
LZNC Anode	Areal capacity (mA h cm ⁻²)	14.87
	Thickness (μm)	80
	Stacked number of layers	16
Separator	Areal weight (mg cm ⁻²)	1
Al Foil	Thickness (μm)	12
Electrolyte	Volumes (g)	11.4
	Energy (W h)	32.2924
Cell	Mass (g)	58.35
	Gravimetric Energy Density (Wh kg ⁻¹)	553.43

Comment (4): The LZNC lithium-containing composite prepared in this manuscript exhibits good cycling performance when matched with the lithium-containing NCM811 cathode. However, it remains unclear whether it has been matched with other cathodes, such as fabricating Li||S pouch cells to test its cycling performance with lithium-free cathodes.

Responses to Comment 4: Thank you for this insightful comment. To evaluate the compatibility of the LZNC anode with lithium-free cathodes, we assembled LZNC||S coin cells and Ah-level LZNC||S pouch cells. The pouch cell utilized a conventional CNT-S composite cathode (70% sulfur content, obtained by integrating sulfur and CNTs at 155 °C for 20 h) with an areal capacity of 4.8 mA h cm⁻², and employed a conventional DOL/DME-based electrolyte (1 M LiTFSI). As shown in Fig. R38, the LZNC||S coin cell exhibited stable cycling over 200 cycles at 0.5C, while the Ah-level pouch cell achieved 60 stable cycles at 0.1C. It should be noted that the cycling stability

of lithium-sulfur batteries is not solely influenced by the anode, but is significantly influenced by the dissolution of polysulfides at the cathode. Future work will focus on optimizing both the sulfur-carbon composite cathode and the electrolyte to further improve cycling performance. These results confirm that the LZNC anode also delivers excellent performance when paired with a lithium-free cathode, highlighting the universal stability of this composite anode.

In the revised manuscript, we have included more discussion in the text and Supplementary materials to clarify the point (*Text of Fig. S41*) and updated the manuscript accordingly (*Page 14 and 16*).

Fig. R38. Cycling performance and the corresponding voltage profiles of LZNC||S coin-type full cell and pouch cell.

Comment (5): In Figure 1c, the scale bars for the 8 μm thickness test are misaligned. Please carefully check and correct this.

Responses to Comment 5: We thank the reviewer for identifying this oversight. The scale bars in *Figure 1c* corresponding to the 8 μm thickness have been carefully realigned and corrected in the revised manuscript.

Fig. R39. Cross-sectional SEM images of LZNC with varying thicknesses.

Comment (6): In Figure S22, for the LZNC symmetric cell, the polarization voltage increases significantly at around 150 h of cycling and then decreases slowly. It is necessary to clarify the underlying mechanism for this phenomenon

Responses to Comment 6: Thank you for pointing out this observation. The transient increase in polarization voltage around 150 h in the LZNC symmetric cell is likely attributed to slight ambient temperature variations or sporadic, localized micro-short circuits during cycling, both of which can temporarily influence ionic transport and interfacial kinetics. To rigorously exclude such potential artifacts, we repeated the cycling test under stringent, controlled conditions. The resulting data (Fig. R40) show a consistently stable polarization profile, confirming that these transient fluctuations do not affect the overall long-term electrochemical stability or the low-polarization characteristic of the LZNC symmetric cells.

In the revised manuscript, we have updated the latest data in the supplementary materials (*Text of Fig. S28*).

Fig. R40. Voltage profile of symmetric cell with LZNC under 5 mA cm^{-2} with a capacity of 5 mA h cm^{-2} .

Comment (7): In Figure 4c, it is suggested that different current densities should be distinguished, and the corresponding current density values should be indicated.

Responses to Comment 7: We thank the reviewer for this suggestion. The current density values have been clearly indicated on Figure 4c to distinguish the different rates, which helps to enhance the clarity of the presentation (see updated Fig. 4c).

Fig. R41. Rate performances of bare Li and LZNC symmetric cells at the current densities from 0.5 to 10 mA cm⁻² with 1 mA h cm⁻².

Comment (8): Concerning Figures 6a and 6f, it is recommended that the corresponding N/P ratios and cathode loadings be labeled in the figures to ensure data completeness.

Responses to Comment 8: We thank the reviewer for this valuable suggestion. In response, we have updated Figures 6a and 6f to explicitly include the corresponding N/P ratios and cathode areal loadings, which ensures all critical parameters are explicitly stated for clarity (see updated Fig. 6a and 6f).

Fig. R42. a) Cycling performance of NCM811||Li and NCM811||LZNC coin-type full

cells at 1C. b) Cycling performance of NCM811||LZNC pouch cells at 0.5C.

Comment (9): Figure 2a presents the XRD data of the LZNC composite, but there is an absence of peaks corresponding to lithiated carbon nanotubes. A reasonable explanation for this should be provided.

Responses to Comment 9: We thank the reviewer for this insightful observation. The absence of distinct lithiated carbon nanotube (LiC_x) diffraction peaks in the XRD pattern of the LZNC composite (Figure 2a) can be attributed to its low crystallinity, wherein the relatively weak scattering signal from the carbonaceous component is masked by the dominating scattering intensity from the highly crystalline metallic and alloy phases within the composite.

As detailed in our *Response to Comment 1*, LiC_x is indeed formed, as confirmed by XPS and the characteristic shift of the (002) graphite peak. However, its XRD signals are broad and weak due to the disordered structure and nanoscale dimensions of the CNTs, which result in low intensity. This is consistent with reports in the literature, such as that by Tilmann et al. [32], who also observed broad, low-intensity features in XRD patterns of lithiated CNTs produced via melt-infusion process.

In the revised manuscript, we have included more discussion in the text and Supplementary materials to clarify the point (*Text of Fig. S13*) and updated the manuscript accordingly (*Page 6*).

Fig. R43. a) XRD pattern of LZNC. b) XRD patterns of pristine CNTs and LZNC, showing an enlarged view of the region corresponding to the dashed rectangle in (a).

References

1. Luo, C. *et al.* Roll-to-roll fabrication of zero-volume-expansion lithium-composite anodes to realize high-energy-density flexible and stable lithium-metal batteries. *Adv. Mater.* **34**, 2205677 (2022).
2. Xie, J. *et al.* Incorporating flexibility into stiffness: self-grown carbon nanotubes in melamine sponges enable a lithium-metal-anode capacity of 15 mA h cm⁻² Cyclable at 15 mA cm⁻². *Adv. Mater.* **31**, 1805654 (2019).
3. Liu, L. *et al.* Free-standing hollow carbon fibers as high-capacity containers for stable lithium metal anodes. *Joule* **1**, 563–575 (2017).
4. Zhang, R. *et al.* Coralloid carbon fiber-based composite lithium anode for robust lithium metal batteries. *Joule* **2**, 764–777 (2018).
5. Xue, P. *et al.* A hierarchical silver-nanowire–graphene host enabling ultrahigh rates and superior long-term cycling of lithium-metal composite anodes. *Adv. Mater.* **30**, 1804165 (2018).
6. Shi, P. *et al.* Inhibiting intercrystalline reactions of anode with electrolytes for long-cycling lithium batteries. *Sci. Adv.* **8**, eabq3445 (2022).
7. Wan, M. *et al.* Mechanical rolling formation of interpenetrated lithium metal/lithium tin alloy foil for ultrahigh-rate battery anode. *Nat. Commun.* **11**, 829 (2020).
8. Qing, P. *et al.* Highly reversible lithium metal anode enabled by 3D lithiophilic–lithiophobic dual-skeletons. *Adv. Mater.* **35**, 2211203 (2023).
9. Li, G. *et al.* Locking active Li metal through localized redistribution of fluoride enabling stable Li-metal batteries. *Adv. Mater.* **35**, 2207310 (2023).
10. Fu, X., Duan, H., Zhang, L., Hu, Y. & Deng, Y. A 3D Framework with an In Situ Generated Li₃N Solid Electrolyte Interphase for Superior Lithium Metal Batteries. *Adv. Funct. Mater.* **33**, 2308022 (2023).
11. Liang, H. *et al.* Boosting the Intrinsic Stability of Lithium Metal Anodes by an Electrochemically Active Encapsulating Framework. *Adv. Energy Mater.* **13**, 2302755 (2023).
12. Zhang, Y., Yao, M., Wang, T., Wu, H. & Zhang, Y. A 3D Hierarchical Host with Gradient-Distributed Dielectric Properties toward Dendrite-free Lithium Metal Anode. *Angew. Chem. Int. Ed.* **63**, e202403399 (2024).
13. Liu, X. *et al.* Constructing Fast Ion/Electron Conducting Pathway within 3D Stable Scaffold for Dendrite-Free Lithium Metal Anode. *Adv. Funct. Mater.* **35**, 2420382

(2025).

14. Zhang, X. *et al.* Topology Fortified Anodes Powered High-Energy All-Solid-State Lithium Batteries. *Adv. Mater.* **37**, 2506298 (2025).
15. Shi, P. *et al.* A Successive conversion-deintercalation delithiation mechanism for practical composite lithium anodes. *J. Am. Chem. Soc.* **144**, 212–218 (2022).
16. Guo, J. *et al.* A self-Reconfigured, dual-Layered artificial Interphase toward high-current-density quasi-solid-state lithium metal batteries. *Adv. Mater.* **35**, 2300350 (2023).
17. Xu, Q. *et al.* Li₂ZrF₆-based electrolytes for durable lithium metal batteries. *Nature* **637**, 339–346 (2025).
18. Zhang, S. *et al.* Oscillatory solvation chemistry for a 500 Wh kg⁻¹ Li-metal pouch cell. *Nat. Energy.* **9**, 1285–1296 (2024).
19. Niu, C. *et al.* High-energy lithium metal pouch cells with limited anode swelling and long stable cycles. *Nat. Energy.* **4**, 551–559 (2019).
20. Zhang, Q.-K. *et al.* Homogeneous and mechanically stable solid–electrolyte interphase enabled by trioxane-modulated electrolytes for lithium metal batteries. *Nat. Energy.* **8**, 725–735 (2023).
21. Ye, Y. *et al.* Solid-solution or intermetallic compounds: phase dependence of the Li-alloying reactions for Li-metal batteries. *J. Am. Chem. Soc.* **145**, 24775–24784 (2023).
22. Huang, K. *et al.* Regulation of SEI formation by anion receptors to achieve ultra-stable lithium-metal batteries. *Angew. Chem. Int. Ed.* **60**, 19232–19240 (2021).
23. Peng, J. *et al.* Tailoring Solvation Structures via Precise Diluent Engineering for High-Rate 500 Wh kg⁻¹ Lithium-Metal Batteries. *Adv. Mater.* e09109 (2025).
24. Ji, H. *et al.* Liquid–liquid interfacial tension stabilized Li-metal batteries. *Nature* **643**, 1255–1262 (2025).
25. Popova, A. N. Crystallographic analysis of graphite by X-Ray diffraction. *Coke Chem.* **60**, 361–365 (2017).
26. Warren, B. E. X-Ray Diffraction in Random Layer Lattices. *Phys. Rev.* **59**, 693–698 (1941).
27. Schweidler, S. *et al.* Volume Changes of Graphite Anodes Revisited: A Combined Operando X-ray Diffraction and In Situ Pressure Analysis Study. *J. Phys. Chem. C* **122**, 8829–8835 (2018).

28. Petkov, V., Timmons, A., Camardese, J. & Ren, Y. Li insertion in ball-milled graphitic carbon studied by total x-ray diffraction. *J. Phys.: Condens. Matter* **23**, 435003 (2011).
29. Shao, Y. *et al.* Drawing a Soft Interface: An Effective Interfacial Modification Strategy for Garnet-Type Solid-State Li Batteries. *ACS Energy Lett.* **3**, 1212–1218 (2018).
30. Duan, J. *et al.* Is graphite lithiophobic or lithiophilic? *Natl Sci Rev* **7**, 1208–1217 (2020).
31. Futaba, D. N., Yamada, T., Kobashi, K., Yumura, M. & Hata, K. Macroscopic Wall Number Analysis of Single-Walled, Double-Walled, and Few-Walled Carbon Nanotubes by X-ray Diffraction. *J. Am. Chem. Soc.* **133**, 5716–5719 (2011).
32. Fuchs, T. *et al.* Increasing the Pressure-Free Stripping Capacity of the Lithium Metal Anode in Solid-State-Batteries by Carbon Nanotubes. *Adv. Energy Mater.* **12**, 2201125 (2022).